# Viewing low back pain through the lens of spinal evolution: Understanding the morphology and limits of the human spine

Abdullah Emre Taçyıldız[1‡*], Özden Erhan Sofuoğlu[2‡], Aydın Sinan Apaydın[1], Melih Üçer[3]

1 Department of Neurosurgery, Karabuk University Faculty of Medicine, Karabuk, Turkey, 2 Department of Neurosurgery, Bakırköy Psychiatric and Neurological Diseases Research and Education Hospital, İstanbul, Turkey, 3 Department of Neurosurgery, Biruni University Faculty of Medicine, Istanbul, Turkey

‡ These authors contributed equally to this work.
* abdullahemretacyildiz@gmail.com

## Abstract

### Background and objective

Low back pain remains a pervasive global health challenge, with significant disability and socio-economic burden. While contemporary biomechanical and occupational factors are well-studied, the role of human spinal evolution and its divergence from modern postural behaviors is less frequently examined. This study aims to *visually explore* and illustrate the historical evolution of human spinal posture through artistic representations, conceptually highlighting the potential biomechanical mismatch between our spine's evolutionary adaptations and current lifestyle-driven postures.

### Methods

We conducted a *qualitative visual analysis* of human figures depicted in selected artworks from three distinct historical periods: the hunter-gatherer era, the agricultural transition, and the post-industrial age. Observed spinal postures were *qualitatively compared* to established biomechanical data on intradiscal pressure levels, derived from previous in-vivo studies. This comparison was used to *illustrate potential physiological or pathological loading* on the spine across different historical contexts.

### Results

Our visual observations suggest a noticeable shift in depicted human postures over time. Figures from the hunter-gatherer period primarily exhibit upright, dynamic positions with an apparent absence of prolonged sitting or significant forward flexion. In contrast, artworks from agricultural and post-industrial societies frequently portray individuals in more flexed, static, and often ergonomically suboptimal postures, including prolonged sitting, bending, and heavy lifting with improper form. These

**Data availability statement:** All data underlying the findings reported in this study are fully available within the article.

**Funding:** The author(s) received no specific funding for this work.

**Competing interests:** The authors have declared that no competing interests exist.

observed postural trends visually align with positions independently associated with increased intradiscal pressures and greater spinal strain in biomechanical literature.

## Conclusion

This study visually traces the evolution of human spinal posture from the hunter-gatherer era to modern industrial life, highlighting a shift from dynamic, biomechanically healthy positions to static and suboptimal postures. These changes, reflected in historical art and linked to lifestyle transitions such as agriculture and industrialization, may underlie the rising prevalence of spinal disorders. The findings suggest that aligning modern practices with the spine's evolutionary design could help prevent and manage spinal pathologies.

## Introduction

Low back pain (LBP) is a widespread global health problem, affecting nearly all individuals at some point in their lives and contributing to significant social, economic, and functional burdens worldwide [1]. The prevalence of LBP is remarkably high, with a notably small number of individuals reporting never experiencing this condition in their lifetime [2–4]. While various risk factors have been identified, an often-underexplored aspect is the potential biomechanical mismatch between the spine's evolutionary structure and the demands imposed by modern life [1–5]. Contemporary habits, such as prolonged sitting, reduced physical activity, and non-physiological postures, deviate considerably from the adaptive patterns for which the human spine evolved, leading to increased mechanical stress on spinal structures [5]. As articulated by Lovejoy, a misalignment between modern human behaviors, occupational demands, and the evolutionary design of our locomotor system may contribute to widespread spinal dysfunction [6].

Historically, two major societal transitions have profoundly influenced human spinal biomechanics. The first was the shift from a nomadic, hunter-gatherer existence to more sedentary agricultural societies. This transition introduced novel forms of physical activity, often characterized by repetitive, labor-intensive tasks and prolonged static postures, which in turn influenced bone density, joint health, and overall biomechanics [7,8]. The second significant transformation occurred during the Industrial Revolution, further intensifying sedentary behaviors, mechanizing labor, and often creating poor ergonomic working conditions [9–11]. These profound societal changes coincided with increased spinal strain, repetitive tasks, and urbanization, all contributing to a discernible rise in spinal and musculoskeletal disorders [9–11].

Early perspectives, such as Krogman's, attributed the origin of low back pain partly to the inherent biomechanical challenges of bipedalism [12]. However, Putz and Müller-Gerbl proposed an alternative viewpoint, suggesting that spinal disorders are not inherently linked to bipedalism itself, but rather to increased lifespan and contemporary lifestyle changes that push beyond the spine's natural adaptive capacity [5].

We align with this latter perspective, positing that the human spine's evolutionary adaptation to bipedalism is fundamentally effective for its intended function. Nevertheless, contemporary spinal disorders appear to arise largely from lifestyle-related postural deviations that fall outside the spine's optimal biomechanical design parameters. A limited understanding of spinal morphology and its movement constraints often exacerbates these issues, particularly in the context of the sedentary behaviors that emerged with the agricultural transition and were further intensified by the Industrial Revolution. In this study, we aim to visually explore how changes in daily life—from prehistoric times through the agricultural era to the post-industrial period—have influenced human spinal posture. By examining anatomical positions as depicted in a selection of cave paintings, sculptures, and classical artworks, we seek to illustrate the conceptual evolutionary and biomechanical disconnect between the spine's original design and the postural demands imposed by modern living. Understanding human gait biomechanics, as emphasized by Lovejoy, is not only crucial for evolutionary insights but also holds practical applications in fields such as implant design and surgical planning [6].

## Methods

### Visual data selection

To visually explore the evolution of human spinal posture across different historical periods, a diverse selection of artworks depicting human figures was purposively chosen. These visual materials were categorized into three distinct historical periods for analysis: the hunter-gatherer era, the transition to settled agricultural life, and the post-industrial period. The selection aimed to provide illustrative examples of postures prevalent during these transformative stages, focusing on how daily life activities might be reflected in anatomical positions. All visual materials included in this study are either covered under Creative Commons licenses or classified as public domain, with full references provided for proper attribution. Specifically, six cave paintings were selected to represent the hunter-gatherer period, while an additional six images were used to illustrate agricultural and industrial transitions

These artworks were categorized into three chronological groups:

- Hunter-Gatherer Era (Fig 1A–1F) – prehistoric cave paintings representing early human movement and posture;

- Agricultural Transition (Figs 2A–2F and 3A–3F) – artworks from ancient Egyptian and early agrarian societies showing seated or flexed working postures;

- Post-Industrial Period (Figs 4A–4E, 5A–5E, and 6A–6E) – depictions of industrial laborers in factory and workshop settings.

All images were sourced from open-access or public-domain repositories (as detailed in the Figure Legends) and selected to illustrate common daily postures within each historical context.

The spinal alignments of the depicted figures were analyzed qualitatively, focusing on the curvature of the lumbar and thoracic regions, the degree of flexion or extension, and the nature of the depicted activity.

In Fig 1A–1F, the observed postures predominantly show upright and dynamic movement patterns. In contrast, Figs 2A–2F and 3A–3F include numerous examples of forward flexion, seated work, and load handling with bent spines.

The industrial artworks (Figs 4A–4E, 5A–5E, and 6A–6E) portray seated, leaning, or weight-bearing postures that imply sustained spinal loading.

Each of these visual observations was conceptually aligned with previously published in-vivo measurements of intradiscal pressures by Wilke et al. [13] and Nachemson [14].

Physiological postures such as upright standing and supported sitting were associated with lower intradiscal pressures, whereas non-physiological positions—unsupported sitting, trunk flexion beyond 60°, or lifting from a bent position—correspond to pathological load levels (as illustrated in Figs 2–5). This alignment was descriptive and comparative in nature,

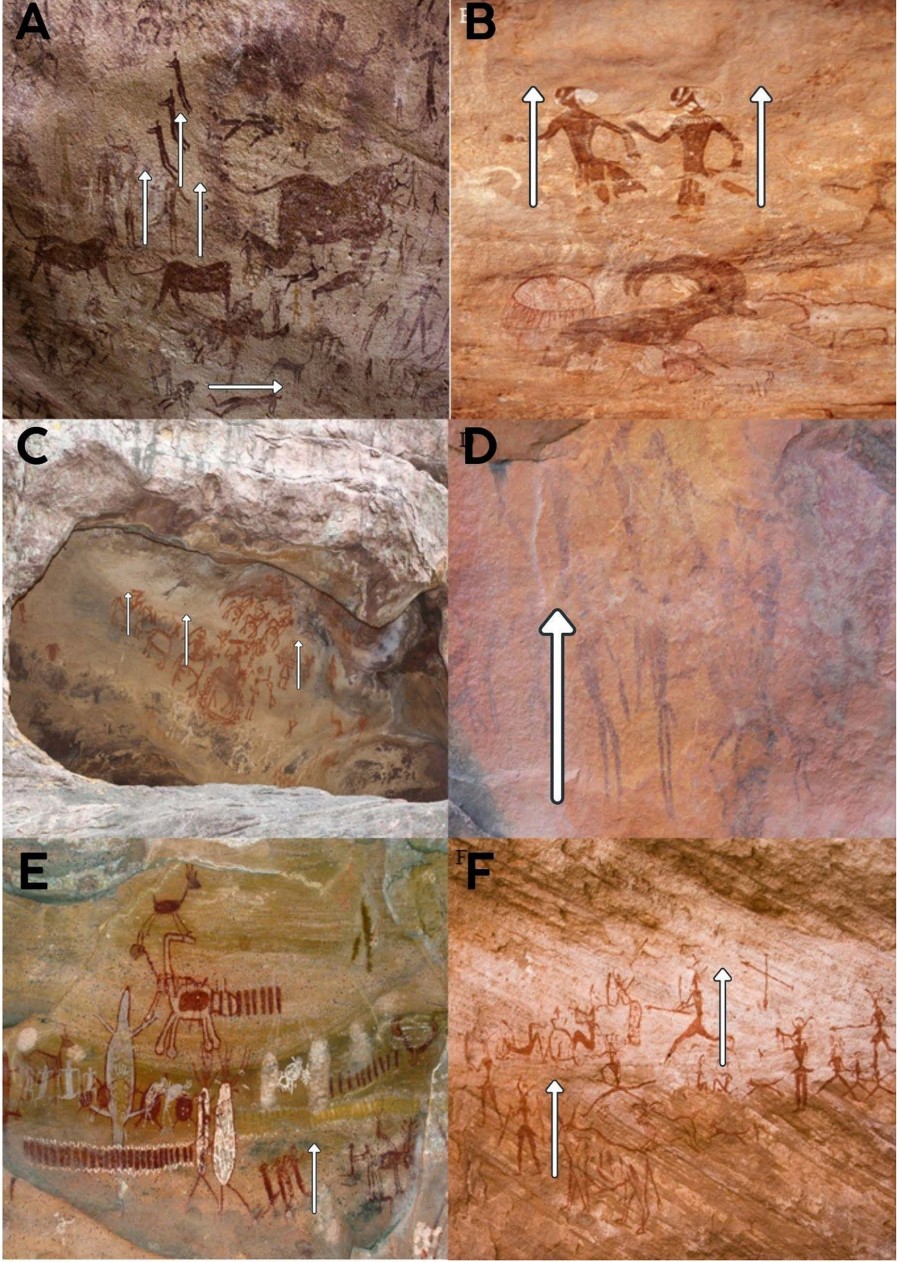

**Fig 1. Evolution of human spinal posture from prehistoric to modern times. A-F**: In the observed cave paintings, figures from the hunter-gatherer period are depicted with upright spines. Positions such as sitting, leaning forward while sitting, bending forward, and lifting weights from a bent position are not observed. References: 1A: User: Clemens Schmillen. Title: Rock paintings from the Cave of Beasts (Gilf Kebir, Libyan Desert) Estimated 7000 BP. (Accessed: 13.02.2024, https://commons.wikimedia.org/wiki/File:Bestias11.JPG) 1B: User: Alessandro Passare, Title: Round Head figures and zoomorphic figures, including a Barbary sheep (Accessed: 13.02.2024, https://commons.wikimedia.org/wiki/File:Fondazione_Passar%C3%A9_V1_056.jpg) 1C: User: Bernard Gagnon, Title: Rock Shelters of Bhimbetka (Accessed: 12.02.2024, https://commons.wikimedia.org/wiki/File:Rock_Shelter_8,_Bhimbetka_02.jpg) 1D: User: Jimbfleak, Title: San rock paintings from the Western Cape in South Africa. (Accessed: 13.02.2024, https://commons.wikimedia.org/wiki/File:Southafrica468bushman.jpg) 1E: User: Ricardo Andre Frantz, Title: Cave painting at Serra da Capivara National Park, Brazil. (Accessed: 10.02.2024, https://commons.wikimedia.org/wiki/File:Serra_da_Capivara_-_Several_Paintings_2b.jpg) 1F: User: Laca Galuzzi Title: Rock paintings in Tadrart Acacus region of Libya dated from 12,000 BC to 100 AD. (Accessed: 09.02.2024, https://en.m.wikipedia.org/wiki/File:Libya_4924_Pictograms_Tadrart_Acacus_Luca_Galuzzi_2007_cropped.jpg.

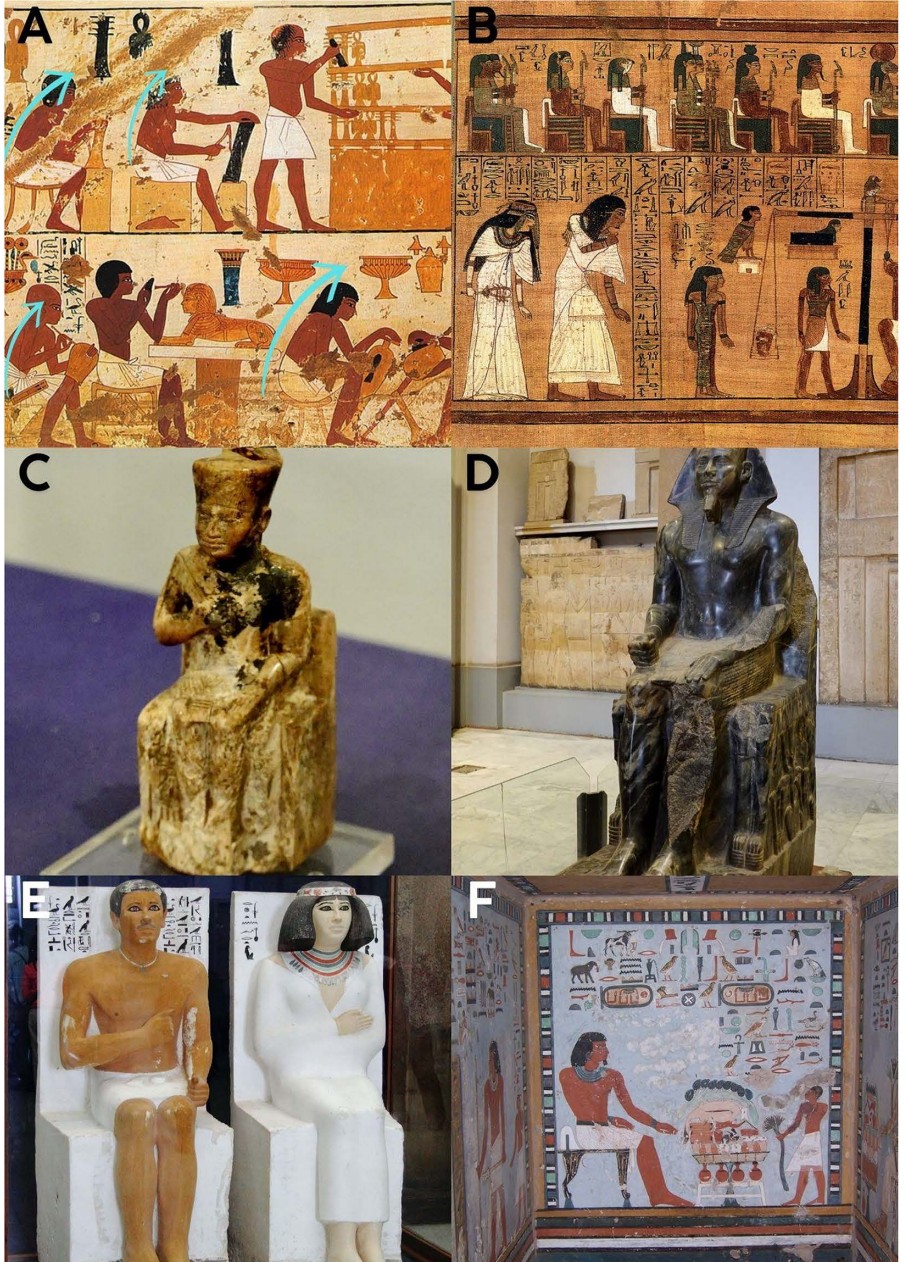

**Fig 2. Early artistic depictions reflecting spinal alignment and load-bearing posture. A-F**: In a painting from ancient Egypt, workers are observed working seated or in flexion. **References 2A**: Artist: Norman de Garis Davies Title: Craftsmen, Tomb of Nebamun and Ipuky, User: Pharos Accessed: 05.02.2024, https://commons.wikimedia.org/wiki/File:Craftsmen,_Tomb_of_Nebamun_and_Ipuky_MET_eg30.4.103a.jpg **2B:** Author: Photographed by the British Museum; original artist unknown Date: Photograph published 2001; artwork created c. 1300 BC User: A. Parrot Accessed: 01.03.2025, https://tr.wikipedia.org/wiki/M%C4%B1s%C4%B1r#/media/Dosya:BD_Weighing_of_the_Heart.jpg **2C:** Author: Olaf Tausch Date: 16 October 2019 Source: Own work User: Neoclassicism and Enthusiast Accessed: 01.03.2025, https://upload.wikimedia.org/wikipedia/commons/f/f7/Kairo_Museum_Statuette_Cheops_03_%28cropped%29.jpg **2D:** Author: Olaf Tausch, Date: 16 October 2019, Source: Own work, User: Oltau, Accessed: 01.03.2025, https://upload.wikimedia.org/wikipedia/commons/a/a8/Kairo_Museum_Sitzstatue_Chephren_06.jpg **2E:** Author: Djehouty, Date: 29 March 2016, Source: Own work, User: Djehouty Accessed: 01.03.2025, https://upload.wikimedia.org/wikipedia/commons/4/41/%C3%84gyptisches_Museum_Kairo_2016-03-29_Rahotep_Nofret_01.jpg **2F:** Author: Daniel Csörföly, Date: 2007 January-February, Source: Photos taken by Daniel Csörföly, User: Csörföly and D Accessed: 01.03.2025, https://upload.wikimedia.org/wikipedia/commons/2/26/Aswan%2C_Egypt_WestBankTombs_2007jan15._14_byDanielCsorfoly.JPG.

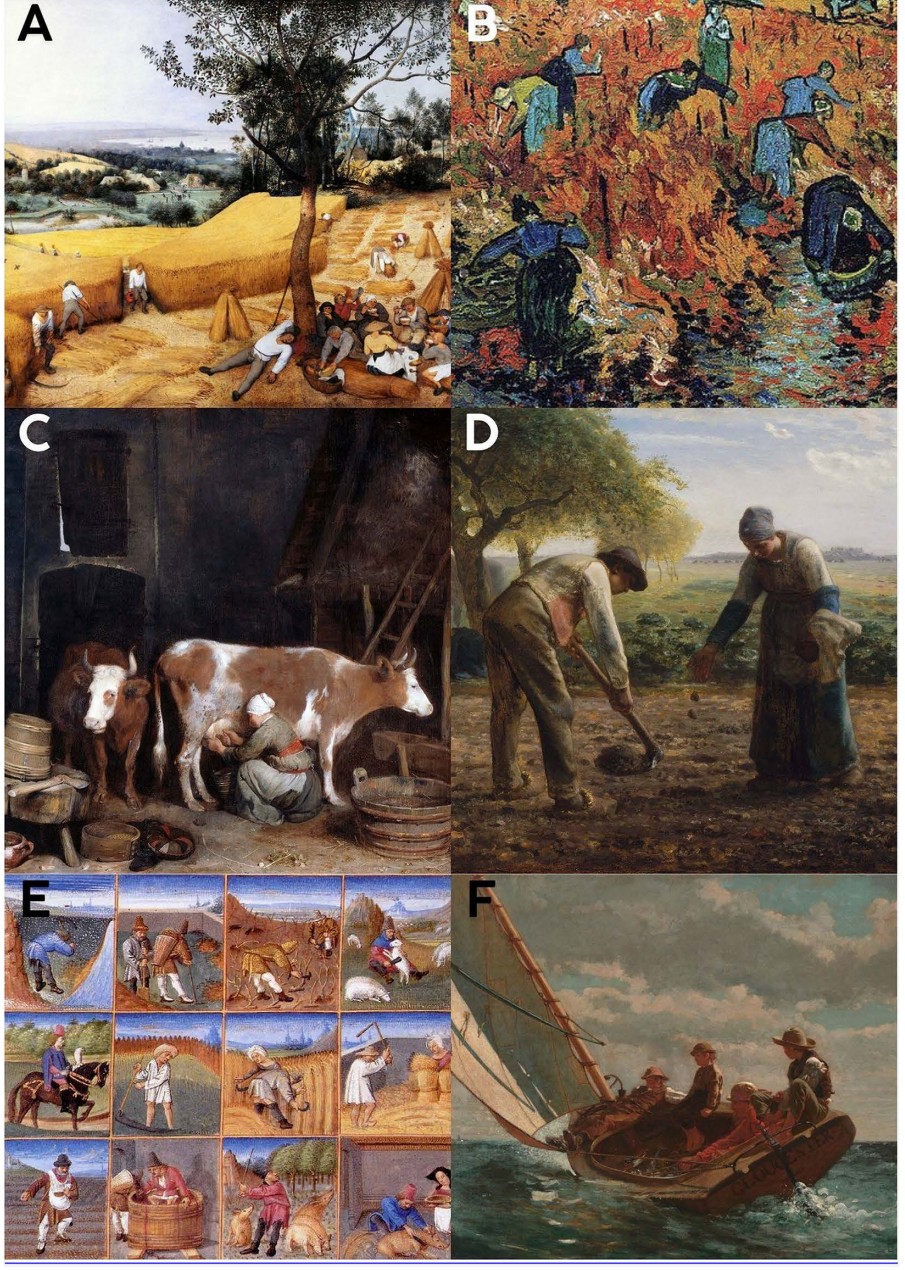

**Fig 3. Classical and Renaissance representations of spinal form and balance. A-F**: Farmers are observed working bent over, lifting objects from the ground while bent over, and working in improper positions. **3 A**: Title: The Harvesters Artists: Pieter Brueghel the Elder, 1565. User: Rolf Kranz, (This work is in the public domain in its country of origin and other countries and areas where the copyright term is the author's life plus 100 years or fewer.) (This work is in the public domain in the United States because it was published (or registered with the **U.**S. Copyright Office) before January 1, 1929.) Accessed: 13.02.2024, https://commons.wikimedia.org/wiki/File:Pieter_Bruegel_the_Elder-_The_Harvesters_-_Google_Art_Project.jpg. **3 B**: Title: The Red Vineyard/ Red Vineyard at Arles. Artist: Vincent van Gogh. User: Coldcreation. This work is in the public domain in its country of origin and other countries and areas where the copyright term is the author's life plus 100 years or fewer. Accessed: 10.02.2024, https://commons.wikimedia.org/wiki/File:Red_vineyards.jpg **3C**: Title: A Maid Milking a Cow in a Barn, Artist: Gerard ter Borch, c. 1652−54, User: Onderwijsgek, Accessed: 1.03.2025, https://upload.wikimedia.org/wikipedia/commons/e/e5/Gerard_ter_Borch_%28II%29_%22De_Koestal%22.jpg **3D**: Title: Potato Planters, Artist: Jean-François Millet, User: DcoetzeeBot, Accessed: 1.03.2025, https://upload.wikimedia.org/wikipedia/commons/9/9d/Jean-Fran%C3%A7ois_Millet_-_Potato_Plant-ers_-_Google_Art_Project.jpg **3E**: Title: Monthly calendar of tasks, Artist: Master of the Geneva Boccaccio, User: Il and Dottore, Date: 1470–1475, https://upload.wikimedia.org/wikipedia/commons/1/1b/Crescenzi_calendar.jpg **3F**: Title: Breezing Up (A Fair Wind), Artist: Winslow Homer, Date: between 1873 and 1876, User: Trzęsacz, Accessed: 1.03.2025, https://upload.wikimedia.org/wikipedia/commons/d/d7/Winslow_Homer_-_Breezing_Up_%28A_Fair_Wind%29.jpg.

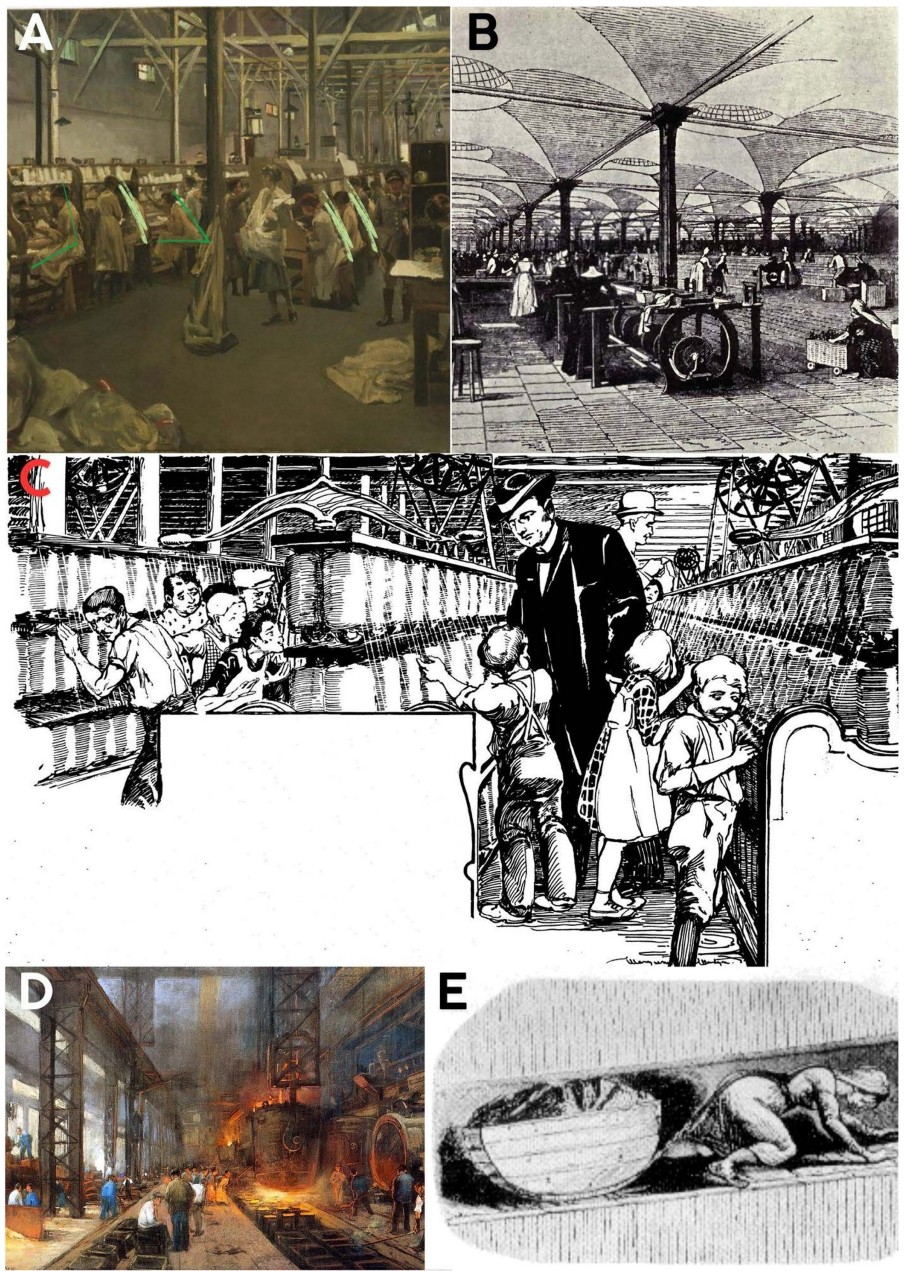

**Fig 4. Industrial-era changes in posture and mechanical loading of the spine. A-E**: Women, men, and children are shown working while seated or leaning forward in a sitting position. References: **4A**: Title: *Army Post Office 3, Boulogne, 1919. Artists*: John Lavery (1856–1941) Users: Rcbutcher. This work is in the public domain in its country of origin and other countries and areas where the copyright term is the author's life plus 70 years or fewer. This work is in the public domain in the United States because it was published (or registered with the U.S. Copyright Office) before January 1, 1929 Accessed: [27.11.2024], https://upload.wikimedia.org/wikipedia/commons/4/4a/Army_Post_Office_3%2C_Boulogne%2C_1919_by_John_Lavery.jpg **4B**: Title: A Day at a Leeds Flax Mill, Artists: No picture credit in book, Date: 1919, original image from 1843, Users: Tagishsimon, Accessed: 04.03.2025, https://upload.wikimedia.org/wikipedia/commons/6/6c/Marshall%27s_flax-mill%2C_Holbeck%2C_Leeds_-_interior_-_c.1800.jpg **4C**: Description: Missouri Governor Joseph W. Folk inspecting child laborers, 1906, drawn by Marguerite Martyn of the *St. Louis Post-Dispatch, Date:* 29 April 1906, Source: Original publication: "St. Louis Post-Dispatch, April 29, 1906, User: BeenAroundAWhile, Accessed: 04.03.2025 https://upload.wikimedia.org/wikipedia/commons/b/b1/Missouri_Governor_Joseph_Folk_inspecting_child_laborers%2C_1906%2C_drawn_by_Marguerite_Martyn.jpg **4D**: Title: The casting of iron in blocks, Artists: Herman Heijenbrock (1871–1948), User: Qlama9 Accessed: 04.03.2025 https://upload.wikimedia.org/wikipedia/commons/f/f0/1890heyenbrock.jpg **4E**: Description: From www.victorianweb.org/history/ashley.html, a educational site offering free info on the victorian age. Image is a copy of one from an official report of a parliamentary commission done in the mid 19th century, Date: 18:41, 11 October 2007, User: Skies, Accessed: 04.03.2025 https://upload.wikimedia.org/wikipedia/commons/7/7b/Coaltub.png.

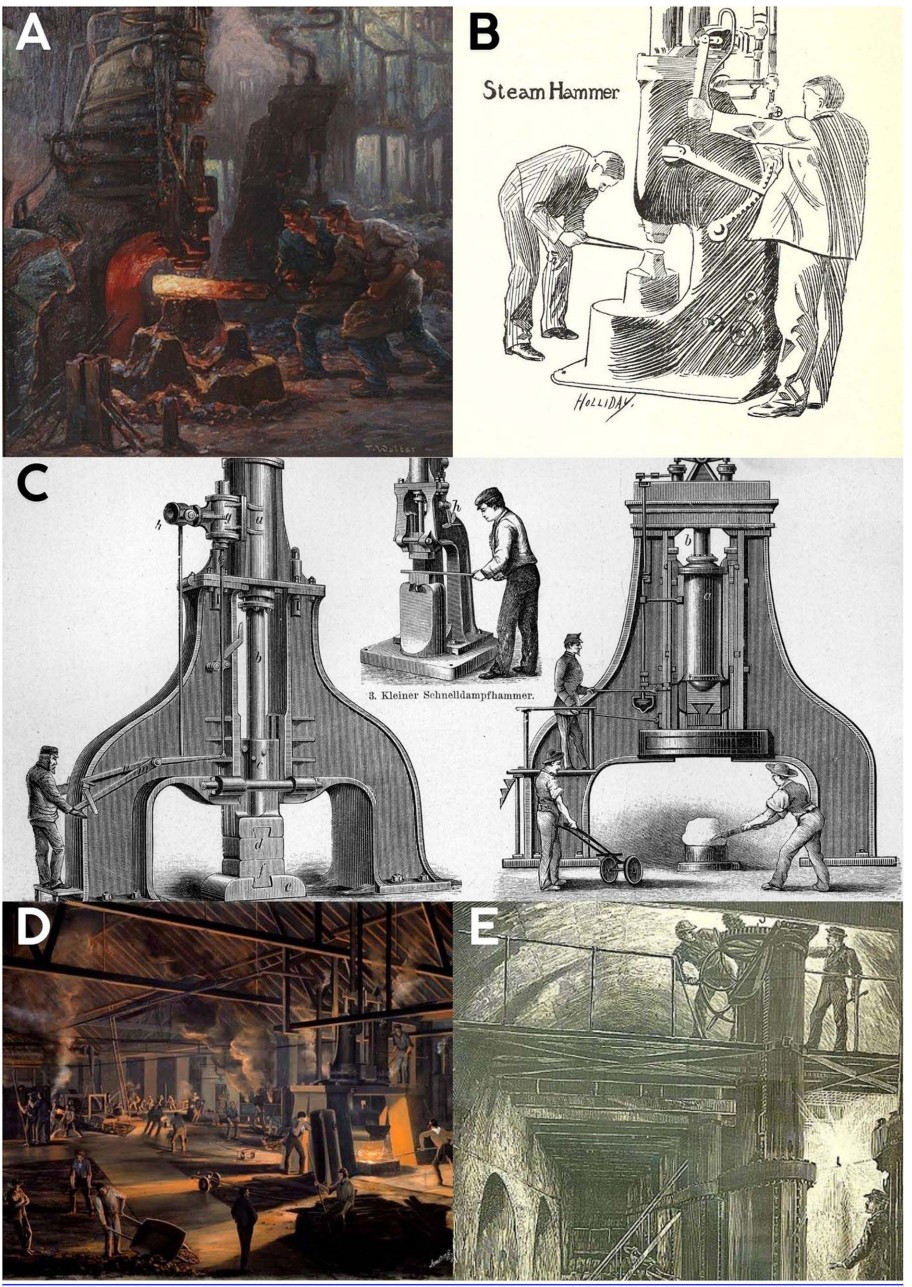

**Fig 5. Modern lifestyle–related postural patterns and spinal stress. A-E**: Factory workers are observed working by bending forward and lifting weights. Reference: **5A:** Title: Painting of factory workers Artist: Toni Anton Wolter. User: Wmpearl (This work was never published prior to January 1, 2003, and is currently in the public domain in the United States) Accessed: 13.02.2024, https://commons.wikimedia.org/wiki/File:Painting_of_factory_workers_by_Toni_Anton_Wolter.jpg **5B:** Title: Annual, 1899 (May), Date: 1899−05; 1899, User: Ssafder, Accessed: 13.02.2024 https://upload.wikimedia.org/wikipedia/commons/f/f5/Annual%2C_1899_%28May%29_-_DPLA_-_14ca89cfa3717f287f4fd0a902420297_%28page_54%29_%28cropped%29.jpg **5C:** Description: Steam engine technology, Date: 1894, Source: Brockhaus' Konversations-Lexikon, 14.Auflage, 4.Band, Author: F.A. Brockhaus, Berlin und Wien, Permission: Author died more than 70 years ago – public domain, User: Hgrobe, Accessed: 04.03.2025, https://upload.wikimedia.org/wikipedia/commons/7/73/Dampfhammer2_brockhaus.jpg **5D**: Author: William Armstrong (1822–1914), Description: English: Painting of the Toronto Rolling Mills, an iron rails factory founded in 1857 by a group of businessmen led by railway magnate Sir Casimir Gzowski. At that time, it was the largest iron mill in Canada and the largest manufacturer in Toronto. The introduction of steel rails led to its closure in 1873., Date: 1864, Source/Photographer: This image is available from the Toronto Public Library under the reference number JRR 1059, User: Skeezix1000, Accessed: 04.03.2025, https://commons.wikimedia.org/wiki/File:Toronto_Rolling_Mills.jpg **5E:** Description: English: Croton Aqueduct shutoff valve, illustration in Harper's Weekly magazine, November 12, 1881, Date: Issue date: November 12, 1881; uploaded May 1, 2008, Author: W. St. John Hasker[?] (difficult to read artist's name (at bottom right — in caption), User: Beyond My Ken, Accessed: 04.03.2025, https://upload.wikimedia.org/wikipedia/commons/d/da/HarpersWeeklyIllustrShuttingOffTheCroton11121881_crop.jpg.

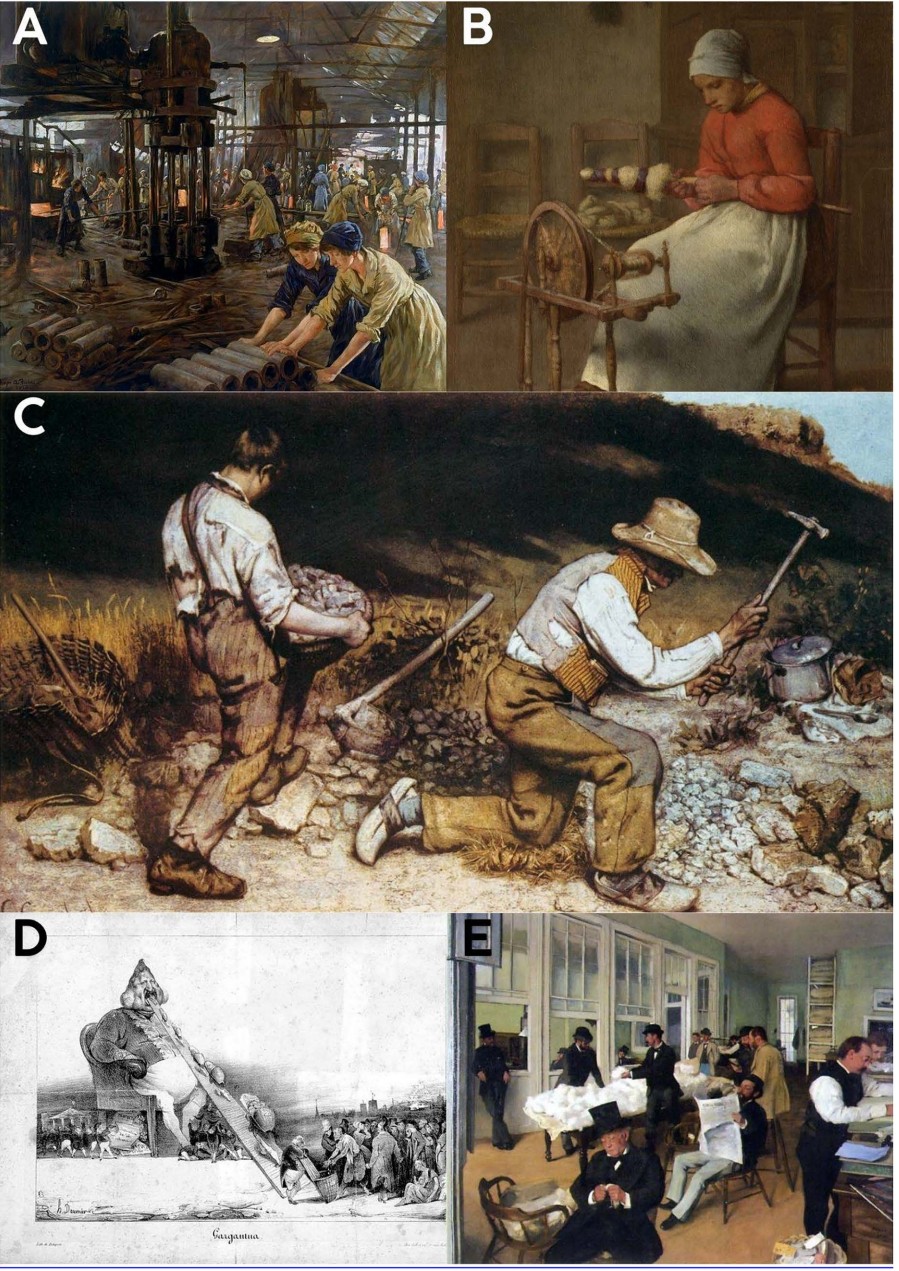

**Fig 6. Improper working postures and ergonomic load on the spine.** A-E: Workers are observed working in improper positions. Reference: **6A:** Title: The Munitions Girls Artist: Stanhope Alexander Forbes) User: Fæ This file is licensed under the Creative Commons Attribution 4.0 International license. Accessed: 16.02.2024 https://en.m.wikipedia.org/wiki/File:%27The_Munitions_Girls%27_oil_painting,_England,_1918_Wellcome_L0059548.jpg **6B:** Title: Woman Spinning, Artist: Jean-François Millet, Date: 1855−186 User: Trzęsacz, Accessed: 04.03.2025, https://commons.wikimedia.org/wiki/File:-Jean-Fran%C3%A7ois_Millet_-_Femme_filature_(1855-60).jpg **6C:** Title: The Stonebreakers, Artist: Gustave Courbet (1819–1877), Date: 1849, User: JarektUploadBot, Accessed: 04.03.2025, https://en.wikipedia.org/wiki/File:Gustave_Courbet_-_The_Stonebreakers_-_WGA05457.jpg **6D:** Description: Gargantua, a lithography by Honoré Daumier, Date: 16 December 1831, Author: Honoré Daumier (1808–1879), User: Racconish, Accessed: 04.03.2025, https://commons.wikimedia.org/wiki/File:Honor%C3%A9_Daumier_-_Gargantua.jpg **6E:** Title: A Cotton Office in New Orleans, Artist: Edgar Degas (1834–1917), Date: 1873, User: SreeBot, Accessed: 04.03.2025, https://commons.wikimedia.org/wiki/File:Cottonexchange1873-Degas.jpg.

designed to contextualize visual postures within a biomechanical framework rather than to derive quantitative values from artworks.

## Qualitative postural observation and conceptual alignment with biomechanical data

The anatomical postures of human figures within the selected artworks were systematically observed and described. This visual analysis involved a qualitative assessment of general spinal alignment, observed ranges of motion (e.g., upright, flexed, seated), and the nature of physical activities depicted.

To conceptually link these visual observations to biomechanical understanding, the observed postures were then qualitatively compared with established intradiscal pressure values. These values, corresponding to various spinal positions, have been quantitatively measured in previous in-vivo studies by Wilke and Nachemson [13,14]. According to their published data, postures such as relaxed standing, lying supine, and supported sitting are considered physiological, associated with lower intradiscal pressures [13,14]. Conversely, postures like unsupported sitting, significant forward flexion, and lifting objects from a bent position are known to be associated with pathological loading and substantially increased intradiscal pressures [13,14]. In this analysis, the general spinal postures observed in the historical artworks were descriptively matched with these broad biomechanical categories to illustrate the potential for physiological or pathological loading as depicted in the art. It is important to note that this process involved a conceptual alignment for illustrative purposes rather than direct quantitative measurement from the artworks themselves.

## Ethical considerations

This study did not require ethical board approval, as it exclusively involves the analysis of publicly available visual materials and does not include research conducted on human participants. All images analyzed are openly licensed under Creative Commons or sourced from the public domain.

## Results

Our visual analysis of selected artworks across different historical periods revealed distinct patterns in depicted human spinal postures. These observations suggest a progressive shift in the types of postures commonly portrayed, moving away from dynamic, upright positions towards more static and flexed ones.

### Hunter-gatherer period (Fig 1A-1F)

Analysis of cave paintings from the hunter-gatherer period consistently demonstrated figures exhibiting predominantly upright spinal postures. Notably, positions such as prolonged sitting, bending forward at the waist, or lifting objects from a bent position were not observed in these depictions. The postures frequently depicted, such as relaxed standing and lying supine, are associated with lower intradiscal pressures, indicating minimal biomechanical strain on the spine according to previous quantitative studies.

### Transition to settled agricultural life (Fig 2A-2F and Fig 3A-3F)

Artworks representing the transition to settled agricultural life, such as those from ancient Egypt (Fig 2A-2F) and depictions of farmers by various artists (Fig 3A-3F), showed a noticeable increase in postures involving sitting and forward flexion. In ancient Egyptian artworks, artisans are frequently shown seated in forward-leaning postures. Similarly, in agricultural scenes, many individuals are depicted bending forward while performing tasks, lifting loads from bent positions, or engaging in work in ergonomically suboptimal flexed postures. These observed postures, particularly those involving significant forward flexion (e.g., maximum flexion sitting, standing bent-forward), are associated with significantly increased intradiscal pressures, potentially contributing to spinal pathologies and accelerated spinal degeneration over time. Manual

handling of loads (MHL) with improper spinal positions was commonly observed among agricultural workers in these depictions.

**Post-industrial age (Fig 4A-4E, Fig 5A-5E, and Fig 6A-6E)**

Following the Industrial Revolution, artworks continued to depict workers in postures associated with significant spinal strain. Figures are frequently shown sitting and bending forward, or carrying loads while bending their spines forward and adopting improper weight-bearing postures. These industrial-era depictions often include both male and female workers engaged in tasks requiring inappropriate spinal positioning. Furthermore, several child laborers are explicitly depicted engaging in tasks or maintaining standing positions with evident spinal misalignment. Such postures are consistent with those linked to high intradiscal pressures, severe mechanical stress, and an increased risk of injury and degenerative conditions in biomechanical literature. The visual evidence from these periods reinforces the prevalence of improper spinal positions, highlighting the physical strain workers experienced.

In summary, the visual evidence across these historical periods indicates a clear progression from predominantly upright and dynamic postures in early human history to more flexed, static, and biomechanically disadvantageous positions in agricultural and industrial societies. These depicted postural shifts align conceptually with established biomechanical understanding of spinal loading and provide a visual correlate to the observed rise in spinal pathologies over time.

## Discussion

Our observational and descriptive study utilized visual representations from various historical periods to illustrate how human spinal postures have changed over time, progressively diverging from the spine's evolutionary design. These transitions, from the nomadic hunter-gatherer period to agricultural and post-industrial societies, reflect fundamental shifts in daily life and corresponding spinal positions (Figs 1–6). We propose that the high global prevalence and incidence of low back pain [15] is significantly influenced by these lifestyle-driven postural changes. Our visual findings and underlying hypothesis align with broader scientific understanding regarding the biomechanical integrity of the spine [5,6].

With the advent of agriculture and settled life, our visual analysis indicates a notable adoption of more stationary and often flexed postures in daily tasks (Figs 2A-2F and 3A-3F). These depictions align with established biomechanical data showing that postures involving significant forward flexion, such as sustained sitting or bending, markedly increase intradiscal pressure and spinal strain [13]. The presence of spinal pathologies in ancient Egyptian mummies, even at relatively young ages, further corroborates the potential link between these early sedentary shifts and spinal health issues [16,17]. Farming occupations, characterized by repetitive bending, lifting, and manual load handling, are indeed associated with high rates of back pain and musculoskeletal disorders [18–20]. Our visual evidence of agricultural work patterns reinforces this association (Fig 3A-3F).

The Industrial Revolution intensified these trends, subjecting workers to prolonged improper spinal positions, heavy lifting, and repetitive movements in factory settings (Figs 4A-4E, 5A-5E and 6A-6E). These occupational demands imposed excessive biomechanical stress on the spine, contributing to a marked increase in back pain and musculoskeletal disorders. [21–23]. The visual data clearly illustrate these physically demanding postures and the resulting strain. Recent data indicating a substantial increase in low back pain prevalence [24] and conditions like knee osteoarthritis [25] in post-industrial society, beyond what can be explained by BMI or longevity alone, further supports our observations. The identification of musculoskeletal disorders in young individuals from historical remains [16,17,25] also aligns with our findings.

Contrary to our hypothesis, some might argue that artwork from the Industrial Revolution period does not necessarily reflect ergonomically sound practices, as artistic interpretations often prioritize aesthetic expression over biomechanical accuracy. However, examining specific works such as *Les Casseurs de pierres* by Gustave Courbet and *Le bigherinaie* by Telemaco Signorini suggests otherwise. In Courbet's painting, the worker on the left bends his knees to support the weight, while the worker on the right adopts a half-kneeling position, a biomechanically effective posture for reducing

spinal loads. Similarly, in Signorini's painting, the weaver on the left maintains a lumbar lordosis, exemplifying proper ergonomic posture during work. Although the women on the right do not adopt the same posture, none of them exhibit a complete loss of lumbar lordosis or a hyperkyphotic stance, further supporting the notion of ergonomically valid depictions in these works. While individuals throughout history undoubtedly demonstrated both correct and incorrect spinal usage, these examples do not undermine the central theme of our study. On the contrary, the broader visual narrative supports the hypothesis that many modern spinal disorders are driven by behavioral and occupational patterns that exceed the biomechanical capacity of the spine's evolutionary design (Figs 1–6) [5,6,26].

However, we believe that such instances, as depicted in various images, do not undermine the central theme of our study.

Conversely, individuals and groups whose lifestyles or occupations demand dynamic movement and varied postures tend to report less back pain. Hunter-gatherer societies exemplify this, with their constant mobility and avoidance of prolonged static or bent positions (Fig 1A-1F) [27,28]. Studies on modern populations also show that incorporating positional changes and regular breaks can significantly reduce back pain risk, even in sedentary office environments [29]. This reinforces the idea that mimicking aspects of our evolutionary postural heritage can be beneficial for spinal health [30].

In conclusion, our study leverages historical artworks to visually demonstrate how humanity has progressively deviated from the natural movements and postures for which the spine evolved [31]. The spine, originally perfected for its intended function, appears to have been undermined by lifestyle changes that impose demands outside its natural design parameters. These shifts have led to a misalignment between the spine's natural structure and the demands of modern living, contributing to increased strain and spinal disorders [5]. Numerous studies from various fields corroborate this overarching hypothesis (Table 1). Ultimately, our findings visually underscore a core tenet of evolutionary theory, as illuminated by the great and respected scientist Charles Darwin: that organisms thrive when their form and function align with environmental demands, and that significant mismatches, as observed in modern human posture, can lead to widespread dysfunction [32].

This study, while offering a visual exploration of postural shifts over time, is subject to several significant methodological limitations that warrant explicit acknowledgment. First and foremost, the primary data source—artworks (including cave paintings, agricultural depictions, and industrial-era images)—are inherently subjective and open to interpretation. Their content may be influenced more by symbolic, stylistic, or ideological factors than by strict biomechanical accuracy. For instance, some industrial-era paintings were intentionally created to emphasize harsh labor conditions or social critiques, which could demonstrably bias postural representations away from reality. This fundamental characteristic of artistic expression makes it challenging, if not impossible, to derive precise or universally representative biomechanical data.

Second, the artworks analyzed constitute a highly selective and non-representative sample of human experiences and postural habits across vast historical periods. The chosen visual data cannot account for the full demographic, cultural, or regional diversity of each historical era. Consequently, any inferences drawn about general population postures or biomechanical trends based solely on these depictions must be viewed with extreme caution and are primarily illustrative rather than statistically generalizable.

Third, numerous critical variables impacting spinal health in past populations—such as specific diet composition, meal frequency, underlying metabolic factors, genetic predispositions, and precise overall life expectancy—could not be directly assessed from the visual data. These elements, while beyond the scope of this visual analysis, are known to significantly influence the development and prevalence of spinal pathologies.

Furthermore, the process of interpreting postural features from artworks is inherently subject to observer bias. As such, despite efforts to apply consistent visual criteria, the absence of standardized inter-rater reliability assessments limits the objectivity and reproducibility of our observations.

Recent myofascial research, including the 'Anatomy Trains' model proposed by Myers et al. and the fascial continuity studies of Stecco, emphasizes that the spine functions as part of an integrated kinetic chain, even though Stecco's

**Table 1.** Articles supporting our study on the longitudinal changes in spinal posture over time.

| Author | Key Findings of the Author | Comment |
|---|---|---|
| Vos T | Low back pain is the most common condition in the population and a leading cause of workforce disability. | This condition affects the majority of the population; however, a global initiative is necessary to drive meaningful change. |
| Putz and Müller-Gerbl | Lifestyle modifications are identified as a prevalent contributing factor to low back pain. | This viewpoint is directly parallel to our hypothesis, emphasizing that the issue stems from lifestyle changes rather than bipedalism. |
| Lovejoy CO | Lovejoy argues that human lifestyles do not align with the evolutionary adaptations of the spine for most of our lives. | Lovejoy's perspectives strongly correspond with the core focus of our study. |
| Wilke and Nachemson | According to their findings, postures such as standing upright, lying down, and sitting with proper support are considered physiological. In contrast, positions like sitting upright, sitting while leaning forward, bending forward, and lifting a load from a bent position are classified as pathological. | Wilke and Nachemson's study demonstrates that intradiscal pressure increases in improper spinal positions, leading to heightened biomechanical stress. This finding directly aligns with the central theme of our study. |
| Pontzer | Comparative studies between hunter-gatherer societies and industrialized populations reveal that hunter-gatherers exhibit significantly greater muscle strength and aerobic capacity. | Pontzer et al.'s study provides direct evidence in support of our hypothesis |
| Fritsch | CT scans of mummies from Ancient Egypt, indicative of a predominantly sedentary lifestyle, have consistently revealed spinal pathologies. Notably, the average age of the analyzed cohort is 38 years. | The occurrence of spinal pathologies in both elite groups and young individuals adapting to a sedentary lifestyle reinforces our findings while contradicting the perspective that spinal disorders are exclusively age-related. Furthermore, this study provides additional validation for our conclusions. |
| Petrella | A separate study on mummies similarly identified a high incidence of spinal pathologies in young individuals. | This study challenges the perspective that low back pain and spinal pathologies are predominantly age-related. Its results, indicative of a sedentary lifestyle, align closely with our research findings. |
| Rosecrance | Farming is globally associated with higher rates of back pain and other musculoskeletal disorders compared to many other occupational groups. | Manual weight carrying and prolonged work in improper postures are significant contributors to low back pain among agricultural and livestock workers. Our visual findings align with those of previous studies. |
| Osborne | Osborn et al. identified lifting, pulling, and pushing as the primary causes of back pain in farming activities | Manual weight carrying and prolonged work in improper postures are significant contributors to low back pain among agricultural and livestock workers. Our visual findings align with those of previous studies. |
| Shivakumar | Manual weight carrying and prolonged work in improper postures are significant contributors to low back pain among agricultural and livestock workers. | Manual weight carrying and prolonged work in improper postures are significant contributors to low back pain among agricultural and livestock workers. Our visual findings align with those of previous studies. |
| Al-Salameen | Workers in paint, steel, automobile, and welding factories exhibit a high prevalence of low back pain.. | The Industrial Revolution led to a growing demand for labor under conditions that necessitated improper spinal postures. The resulting factory environments forced workers into ergonomically unsuitable positions, which contributed to the increased prevalence of low back pain. |
| Burdorf | A comparative analysis of back pain among workers in concrete production plants and engineers within the same facilities demonstrated a significantly higher prevalence of back pain among manual laborers | Individuals engaged in more strenuous labor and prolonged improper postures experienced a higher incidence of back pain. The findings of this comparative study provide additional support for our hypothesis. |
| Yasim | The study revealed that avoiding improper postures is associated with an 80% reduction in the risk of developing back pain. | The reduced prevalence of pain among individuals who adhere to the natural biomechanical constraints and evolutionary design of the spine aligns with our findings and reinforces our hypothesis |
| Wallace | The prevalence of musculoskeletal disorders, such as knee osteoarthritis, was lower during the early industrial period but increased in the post-industrial era. | Following the Industrial Revolution, improper working conditions contributed to a rise in musculoskeletal pain, a trend that is also consistent with our visual findings. |
| Stieglitz J | Today, the risk of lower back pain has increased by 50% compared to 25 years ago. | With ongoing modernization, the incidence and prevalence of pain are increasing. |
| Belinchón-deMiguel | In their study replicating the hunter-gatherer lifestyle, Belinchón-de Miguel et al. linked this lifestyle to reduced pain levels and higher physical activity. | After the study on child laborers, this study stands as the strongest support for our findings and assertions. |

work specifically investigates myofascial continuity in the upper limb rather than the spine itself [33–35]. As shown in whole-body myofascial models by Myers et al., segmental dysfunction can propagate tension to distant regions such as the pelvis, sacroiliac joint, and shoulder girdle, while Stecco's upper-limb fascial continuity studies provide anatomical support for this broader principle of force transmission [33–35]. These mechanisms reinforce our hypothesis that modern postural habits generate systemic biomechanical stress exceeding the spine's evolutionary design (Figs 1-6). From a functional and clinical standpoint, the evolution of human posture cannot be fully understood without integrating the role of the myofascial system. The thoracolumbar fascia (TLF) forms a continuous tension network linking the spine, pelvis, and lower limbs, providing mechanical stability and efficient force transmission during movement [36,37]. Within the evolutionary context of human bipedalism, the thoracolumbar fascia (TLF) functions as a key connective interface linking the trunk and pelvis, contributing to load transfer along the axial chain and stabilizing the spine during upright locomotion. In this framework, Schleip et al. further demonstrated that fascia is not a passive membrane but a contractile tissue: human lumbar fascia contains α-SMA–positive myofibroblasts capable of generating active tension, and subtle stiffness changes in these fascial layers may influence neuromuscular coordination through mechanosensory pathways, thereby integrating with the broader biomechanical adaptations that support human spinal posture [38]. Together, these findings support our central hypothesis that modern postural problems stem less from skeletal alignment alone and more from maladaptive changes within the myofascial system. If the thoracolumbar fascia and its paraspinal–pelvic networks function as contractile, load-responsive tissues prone to densification, stiffness shifts, and neuromuscular modulation, then the sedentary and asymmetric loading patterns of contemporary life may disrupt a system that once stabilized the trunk and coordinated lower-limb biomechanics. In this sense, modern postural dysfunction reflects an evolutionary mismatch between a formerly dynamic fascial architecture and today's mechanically constrained environments.

At a micro-physiological level, prolonged static or flexed postures induce ischemic and hypoxic zones within postural muscles such as the multifidus and quadratus lumborum, facilitating myofascial trigger point (TrP) formation [39]. Myofascial trigger points (TrPs) can produce pain patterns that mimic spinal pathology, and while ultrasound imaging can identify focal myofascial abnormalities associated with these pain generators, clinical evidence also shows that ultrasound-guided trigger-point needling (US-DN) effectively reduces myofascial pain and local neuromuscular hyperactivity, underscoring its value as a targeted therapeutic intervention [40,41]. Together, these findings provide a mechanistic bridge linking evolutionary postural mismatch to the clinical manifestations of chronic low back pain (Figs 1-6).

Ultrasound (US) imaging has increasingly become a critical tool in the diagnosis and treatment of myofascial pain. **Bubnov** first demonstrated the feasibility of *real-time sonographic visualization* of myofascial trigger points and reported **93.3% pain relief** in a clinical series of 91 patients treated with US-guided trigger-point dry needling (US-DN) [42]. Building on these findings, a **comparative clinical study** showed that US-guided needling provides **more accurate localization**, **greater pain reduction**, and **higher treatment responsiveness** than palpation-guided techniques [41]. More recent reports further suggest that US-guided interventions may enhance **muscle function** and **regional mobility** by enabling targeted treatment of fascial dysfunction [43]. Collectively, these data underscore how image-guided approaches translate myofascial biomechanics into precise clinical interventions, reinforcing the connection between fascial anatomy, postural evolution, and modern rehabilitation strategies. These ultrasound-guided myofascial findings support our central hypothesis by showing that the fascial and segmental dysfunctions produced by modern, evolution-mismatched loading patterns are not only measurable but clinically reversible. Thus, US-based interventions provide mechanistic confirmation that contemporary postural strain exceeds the adaptive conditions under which the human spine evolved.

From a practical standpoint, the evolutionary–postural framework highlights the need for interventions that restore dynamic spinal loading and myofascial mobility. Approaches such as dynamic posture training, fascial stretching, and core stabilization can counteract the sedentary, asymmetric loading patterns characteristic of modern environments [36]. In addition, ultrasound-guided dry needling (US-DN) provides a targeted method for reducing fascial densification and

improving segmental control [40,41]. Together, these strategies translate evolutionary principles into clinically actionable rehabilitation pathways.

From a comparative anatomical and behavioral standpoint, non-human primates such as chimpanzees typically adopt flexed-hip, flexed-trunk postures—even when seated—and rely less on sustained lumbar lordosis under axial compression. Developmental and positional studies in chimpanzees suggest that these flexed configurations distribute loads across the pelvis and spine in ways that differ from the extended, lordotic lumbar posture characteristic of habitual human bipedalism [44–47]. In contrast, independent evidence from human biomechanics shows that modern sitting—characterized by posterior pelvic tilt, reduced lumbar lordosis, and static hip flexion—deviates markedly from these evolutionarily conserved resting behaviors. This supports the view that prolonged human sitting is not an evolutionarily optimized posture, but rather a biomechanically compromised one shaped by cultural and occupational demands.

A slight head-down posture is perceived as more cooperative, warm, and non-threatening during interpersonal interactions, suggesting that head tilt modulates social perception across cultures [48]. While this adaptation may serve psychosocial purposes, it simultaneously increases cervical and upper thoracic mechanical load, illustrating the continuing conflict between evolutionary optimization and the postural demands of modern social behavior.

According to the in vivo measurements reported by Wilke et al., intradiscal pressure varies substantially with body posture and physical activity, reflecting the dynamic loading characteristics of the lumbar spine. In a lying supine position, mean disc pressure remains minimal at approximately 0.10 MPa, increasing slightly when lying on the side (0.12 MPa) or prone (0.11 MPa) [13]. When the trunk is extended while prone and supported on the elbows, the pressure rises markedly to about 0.25 MPa, illustrating the load transfer through posterior spinal elements. Everyday actions such as laughing or sneezing in a lateral decubitus position elevate pressures to 0.15 MPa and 0.38 MPa, respectively, while turning movements can transiently peak at 0.70–0.80 MPa. During upright posture, relaxed standing produces around 0.50 MPa, which increases to nearly 1.10 MPa when bending forward, and further to 0.92 MPa during a Valsalva maneuver. Sitting generates variable loading depending on posture: from 0.27 MPa in a slouched position to 0.83 MPa with maximal flexion, while active upright sitting reaches approximately 0.55 MPa. Transitioning from sitting to standing momentarily produces high pressures near 1.10 MPa. Locomotor and lifting activities exert even greater variations. Walking yields 0.53–0.65 MPa, whereas jogging ranges from 0.35 to 0.95 MPa depending on footwear. Ascending or descending stairs causes pressures between 0.30 and 1.20 MPa. Notably, lifting 20 kg with a rounded back can increase intradiscal pressure to 2.30 MPa, compared to 1.70 MPa when proper lifting technique is employed, and 1.10 MPa when the load is held close to the body [13]. Maintaining the same load **60 cm from the chest** produces 1.80 MPa, underscoring the mechanical disadvantage of lever arm extension. Even during nocturnal rest, pressures fluctuate between 0.10 and 0.24 MPa, indicating continuous low-level mechanical activity of the spine during recumbency. These data collectively highlight the remarkable sensitivity of spinal load distribution to posture, movement, and external weight, providing an essential biomechanical foundation for ergonomic recommendations and preventive strategies against lumbar disc degeneration.

Comparative studies reveal that workers in the textile industry experience significantly more back pain than teachers [49]. This disparity is likely due to the physically demanding tasks, repetitive movements, and prolonged improper postures commonly associated with textile work. This result is attributed to the fixed working positions of garment workers, which involve prolonged periods of repetitive tasks and static postures, leading to increased strain on the spine and a higher prevalence of back pain [49]. In the images from the hunter-gatherer period, no improper spinal positions were observed, and the figures clearly demonstrate mobility and dynamic postures (Figs 1 A-1F). In contrast, when comparing workers in concrete production facilities with engineers in the same factory, it was found that engineers experience significantly less back pain [22]. This difference is attributed to the physically demanding tasks and prolonged improper spinal positions required of workers, which are not typically part of engineers' roles. The lower prevalence of back pain among engineers is attributed to their minimal or absent time spent in bent-back positions, reducing strain on the spine compared to workers in physically demanding roles [22]. In cave paintings depicting human figures, individuals are almost never

shown in bent positions, reflecting a natural alignment of the spine (Fig 1A-1F). Similarly, studies indicate that individuals with less than 20 years of experience in the driving profession report less back pain, likely due to reduced cumulative exposure to prolonged sitting and improper spinal postures [50]. Individuals professionally exposed to bending or twisting for less than 10 years also report experiencing less back pain, likely due to reduced cumulative strain on the spine and a shorter duration of exposure to improper spinal positions [50]. People involved in agriculture often work in unsustainable spinal positions, such as bending and twisting, which place significant strain on the spine and increase the risk of musculoskeletal disorders over time [19]. In agricultural labor, inappropriate and unsustainable spinal positions are common (Figs 3A-3F). In a study conducted in France, occupational groups at risk for surgically treated disc herniation among men included company managers in firms with more than ten employees, foremen, police officers, soldiers, and skilled workers [51]. Among women, an increased risk of requiring surgical treatment for disc herniation was observed in administrative staff, public employees, sales personnel, and workers providing direct services to individuals [51]. According to the good practice guidelines proposed by the European COST B13 project, the most commonly reported risk factors in the literature for the development of low back pain include heavy physical workload, frequent bending, twisting, lifting, pushing, or pulling movements, repetitive tasks and prolonged static postures, as well as psychological risk factors such as stress, anxiety, depression, cognitive dysfunction, maladaptive "pain behavior" patterns, low job satisfaction, and psychological tension [52]. Analyses of multiple longitudinal studies have confirmed a strong association between certain occupational postures and the incidence of low back pain. Specifically, activities involving lifting, forward or backward bending, trunk rotation, and exposure to whole-body vibration have been identified as significant risk factors [53,54].

While our central argument emphasizes that modern civilization has introduced postural challenges through sedentary behavior, artificial ergonomics, and diminished proprioceptive feedback, it is equally important to recognize that postural evolution itself is not a unidirectional process of degeneration. Anthropological and biomechanical research shows that posture has historically been a multifactorial adaptive trait shaped by environmental and cultural demands. For instance, studies of traditional African populations (Luo tribe) practicing habitual head-load carrying demonstrated **reduced energetic cost and highly efficient load transfer** during gait, reflecting adaptive strengthening of spinal and paraspinal structures [55, 56]. Women of the Kikuyu tribe, in contrast, carry substantial loads supported by a forehead strap, a practice that frequently leads to the development of a permanent cranial groove [55]. Likewise, comparative anatomical analyses indicate that non-human primates, including chimpanzees, exhibit **distinct lumbar and pelvic adaptations** that provide context-specific mobility and axial function, underscoring that spinal form varies according to ecological pressures [6]. These findings refine our argument by showing that posture has always adapted to its mechanical environment; thus, modern civilization may represent a **mismatch** between our evolved spinal design and the novel loading patterns of contemporary life. In this view, the issue is not evolutionary failure, but the rapid divergence between ancestral biomechanical demands and present-day postural exposures.

Finally, significant modern contributors to spinal disorders, such as rising rates of obesity and changes in overall physical activity levels, were not quantitatively integrated into our model. While we acknowledge their importance, our visual analysis could only broadly comment on the observable reduction in physical activity over time, which supports our central hypothesis regarding the mismatch between spinal evolution and modern lifestyle demands (Table 1). However, the absence of direct measurement for these confounders means their precise influence on the observed trends cannot be definitively quantified within this study.

Given these considerable limitations, the findings of this study should be interpreted as a conceptual and illustrative argument for a potential biomechanical mismatch, rather than definitive empirical evidence of population-level postural changes or their direct causal link to spinal disorders. A key limitation lies in the scarcity of ancient records explicitly describing low back pain, making any inference about its link to poor posture based on artistic representations methodologically indirect and lacking robust evidential support. Future interdisciplinary work integrating biomechanical analysis, ethnographic observation, and historical contextualization may help refine these preliminary insights.

## Conclusion

This study has visually illustrated the evolution of human spinal posture from the hunter-gatherer era to the modern industrialized world, utilizing a visual analysis of selected cave art and classical paintings. Our visual findings suggest a progressive shift away from dynamic, biomechanically favorable positions toward more static, flexed, and ergonomically suboptimal postures.

With the advent of agriculture, sedentary routines, and the demands of industrial labor, postures such as prolonged sitting, forward flexion, and improper lifting techniques have become increasingly prevalent. These observed changes visually correlate with the rising incidence of spinal disorders, aligning with both established biomechanical data and the historical visual evidence presented.

We propose that many modern spinal pathologies stem from a fundamental mismatch between our spine's evolutionary design and the physical demands of contemporary life. Aligning spinal care and occupational practices with the spine's natural biomechanical limits may offer effective strategies for prevention and rehabilitation. Understanding and respecting the evolutionary morphology of the spine is essential for restoring balance between the human body and the modern environment.

Ultimately, the solution may not lie in redesigning the spine, but rather in redesigning how we live around it.

## Supporting information

**S1 File. Location of the Data (Repository).** All data underlying the findings reported in this study are fully available within the article. No additional datasets, images, or external repositories were used.
(DOCX)

## Acknowledgments

The authors would like to express their sincere gratitude to Burçin Taçyıldız for her unwavering support and for providing the time and encouragement that made this study possible. We also acknowledge the use of OpenAI's ChatGPT-4 (https://chat.openai.com, accessed 2 March 2024), which assisted in grammar checking during the manuscript preparation process. Additionally, we thank Canva (https://www.canva.com.tr, accessed 14 and 24 March 2024) for providing the design tools used to create Figs 1–6. All content, results, and interpretations presented in this manuscript are solely the work of the authors.

## Author contributions

**Conceptualization:** Abdullah Emre Taçyıldız, Özden Erhan Sofuoğlu.

**Data curation:** Abdullah Emre Taçyıldız, Özden Erhan Sofuoğlu.

**Formal analysis:** Abdullah Emre Taçyıldız, Özden Erhan Sofuoğlu.

**Investigation:** Abdullah Emre Taçyıldız, Özden Erhan Sofuoğlu, Aydın Sinan Apaydın.

**Methodology:** Abdullah Emre Taçyıldız, Özden Erhan Sofuoğlu, Aydın Sinan Apaydın.

**Project administration:** Abdullah Emre Taçyıldız, Özden Erhan Sofuoğlu.

**Resources:** Abdullah Emre Taçyıldız, Özden Erhan Sofuoğlu, Melih Üçer.

**Software:** Abdullah Emre Taçyıldız, Özden Erhan Sofuoğlu.

**Supervision:** Aydın Sinan Apaydın, Melih Üçer.

**Validation:** Aydın Sinan Apaydın, Melih Üçer.

**Visualization:** Abdullah Emre Taçyıldız, Özden Erhan Sofuoğlu.

**Writing – original draft:** Abdullah Emre Taçyıldız, Özden Erhan Sofuoğlu.

**Writing – review & editing:** Abdullah Emre Taçyıldız, Özden Erhan Sofuoğlu, Aydın Sinan Apaydın.

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
