## [Decision Letter · Decision Letter 0]

13 Oct 2025

Dear Dr. Abdullah Emre Taçyıldız,

Thank you for submitting your manuscript to PLOS ONE. After careful consideration, we feel that it has merit but does not fully meet PLOS ONE’s publication criteria as it currently stands. Therefore, we invite you to submit a revised version of the manuscript that addresses the points raised during the review process.

We look forward to receiving your revised manuscript.

Kind regards,

Holakoo Mohsenifar

Academic Editor

PLOS ONE

Journal Requirements:

4. We note that Figures 1, 2, 3, 4, 5 and 6 in your submission contain copyrighted images. All PLOS content is published under the Creative Commons Attribution License (CC BY 4.0), which means that the manuscript, images, and Supporting Information files will be freely available online, and any third party is permitted to access, download, copy, distribute, and use these materials in any way, even commercially, with proper attribution. For more information, see our copyright guidelines: http://journals.plos.org/plosone/s/licenses-and-copyright.

a. You may seek permission from the original copyright holder of Figures 1, 2, 3, 4, 5 and 6 to publish the content specifically under the CC BY 4.0 license.

Reviewers' comments:

Reviewer's Responses to Questions

**Comments to the Author**

1. Is the manuscript technically sound, and do the data support the conclusions?

Reviewer #1: Yes

Reviewer #2: Yes

Reviewer #3: Partly

2. Has the statistical analysis been performed appropriately and rigorously?

Reviewer #1: N/A

Reviewer #2: N/A

Reviewer #3: Yes

3. Have the authors made all data underlying the findings in their manuscript fully available?

Reviewer #1: Yes

Reviewer #2: Yes

Reviewer #3: Yes

4. Is the manuscript presented in an intelligible fashion and written in standard English?

Reviewer #1: Yes

Reviewer #2: Yes

Reviewer #3: Yes

Reviewer #1: Nice and innovative work! This is a novel and qualitative piece of work that blends science and art to explore how evolution may have influenced the spine and contributed to the progression of back pain in the human race.

Please find my minor comments below:

• Page 4, second paragraph: The letter ‘T’ is missing in the sentence beginning “hese profound societal changes coincided with increased spinal strain, repetitive tasks, and urbanization, all contributing to a discernible rise in spinal and musculoskeletal disorders [9-11].”

• Methods section: Please add figure numbers to the text for clarity, as it is currently difficult to determine which figure is being referenced.

• Discussion section: Some words are unnecessarily bolded; this formatting should be removed.

Reviewer #2: The manuscript presents an interesting and visually engaging concept linking spinal evolution with posture and low back pain. However, to enhance its clinical and biomechanical depth, the authors could integrate the concept of myofascial continuity and muscle chains that maintain postural stability. The current version focuses mainly on skeletal evolution, whereas muscular and fascial adaptations play a crucial role in chronic overload and pain syndromes associated with poor posture.

Fascial and Myofascial Chain Perspective

Consider referencing Thomas W. Myers’ “Anatomy Trains” model or Stecco’s fascial continuity research, which demonstrate how dysfunctions in one segment (e.g., thoracolumbar fascia, multifidus, quadratus lumborum) propagate tension throughout the kinetic chain. Chronic seated or flexed postures typically cause shortening of the hip flexors and thoracolumbar fascia, inhibition of the multifidus, and compensatory overactivity of the quadratus lumborum and upper trapezius—mechanically linking the pelvis, sacroiliac joint (SIJ), and shoulder girdle dysfunctions.

Trigger Points and Postural Overload

Introduce the role of myofascial trigger points (TrPs) as a micro-level consequence of prolonged overload or static posture.

Citing Travell & Simons could strengthen the pathophysiological basis: sustained seated posture → ischemic muscle zones → TrP activation in the multifidus and quadratus lumborum → referred pain mimicking spinal disorders.

The addition of ultrasound (US)-based confirmation of TrPs and US-guided dry needling (US-DN) (Bubnov R., Ultrasound Med Biol, 2011; https://link.springer.com/article/10.1186/1878-5085-3-13) would provide a modern pathophysiological bridge between evolutionary mismatch and clinical manifestations of low back pain.

Comparative Functional Anatomy

The authors could contrast human static sitting posture with that of non-human primates (e.g., chimpanzees), which maintain a flexed hip and mobile lumbar spine position even when seated (see https://doi.org/10.1016/j.applanim.2021.105417;
https://doi.org/10.1016/0163-6383(91)90022-K).

This comparison would show that humans’ prolonged sitting with posterior pelvic tilt and loss of lumbar lordosis is not “evolutionarily optimized,” unlike dynamic squat resting seen in apes.

Additionally, anthropological practices such as carrying loads on the head, common in African populations, naturally maintain spinal alignment through balanced axial loading. Including such examples would illustrate cultural mechanisms that promote postural correction and dynamic equilibrium.

Finally, the authors might briefly acknowledge modern behavioral adaptations—such as the slightly head-down posture that improves social perception but may alter cervical–thoracic load distribution (see https://oa.mg/work/10.1016/j.actpsy.2022.103602)—to highlight ongoing tension between evolutionary optimization and modern psychosocial demands.

Clinical Relevance and Preventive Outlook

Discussing interventions that “re-align” with evolutionary and myofascial principles—dynamic posture, ergonomic squatting, fascial stretching, or core stabilization—would add translational value.

A schematic figure or appendix linking the sequence evolutionary posture → fascial adaptation → trigger point formation → chronic pain loop would help visualize the proposed concept and enhance its educational and preventive potential.

Reviewer #3: This article was an interesting study. This study extrapolated from postures depicted in figurative paintings to contemporary research on spinal load dynamics in corresponding positions. It examined the dichotomy between spinal evolution and modern lifestyle through an artistic lens and was groundbreaking and mind-blowing in its methodology.

This study provided an evidence-based alert regarding biomechanically unsustainable modern habits. However, I think there need add some scientific data in this article.

The results show different positions were associated with intradiscal pressures. If the authors provided some data of intradiscal pressure under different position. These data can help to clearly tell the reader which position were risks of low back pain.

The figure 2 provided paintings from ancient Egypt. But people in some pictures were not workers. This could not show the people’s daily tasks. Although the arts from ancient Egyptian are few. I still suggest the author provide more pictures of workers.

In figure 1-6, please label alphabetic markers in subfigures, like A, B, C, et al.

In ancient records, it limited historical documentation on low back pain, which inferred its association with poor posture through artistic depictions appears methodologically indirect and lacks substantive foundation. Why did the authors not prioritize analyzing direct correlations between occupational postures and lumbar pathology in contemporary populations?

**Do you want your identity to be public for this peer review?** For information about this choice, including consent withdrawal, please see our Privacy Policy

Reviewer #1: No

Reviewer #2: **Yes:**  Rostyslav Bubnov

Reviewer #3: No

---

## [Author Response · Author response to Decision Letter 1]

16 Oct 2025

Dear Scientist and Reviewer 1

We sincerely thank the reviewer for their encouraging and insightful feedback. We are pleased that the reviewer found our work innovative and appreciated its interdisciplinary approach combining science and art. We have carefully addressed all the minor comments as outlined below.

Reviewer Comment 1:

Page 4, second paragraph: The letter ‘T’ is missing in the sentence beginning “hese profound societal changes coincided with increased spinal strain, repetitive tasks, and urbanization, all contributing to a discernible rise in spinal and musculoskeletal disorders [9–11].”

Response 1 :

Thank you for catching this typographical error. We have corrected the missing initial letter, and the sentence now reads:

“These profound societal changes coincided with increased spinal strain, repetitive tasks, and urbanization, all contributing to a discernible rise in spinal and musculoskeletal disorders [9–11].”

Reviewer Comment 2:

Methods section: Please add figure numbers to the text for clarity, as it is currently difficult to determine which figure is being referenced.

Response 2:

We appreciate this valuable suggestion. The Methods section has been thoroughly revised to include explicit figure references throughout the text for improved clarity and reader guidance.

These artworks were categorized into three chronological groups:

• Hunter-Gatherer Era (Figure 1A–F) – prehistoric cave paintings representing early human movement and posture;

• Agricultural Transition (Figures 2A–F and 3A–F) – artworks from ancient Egyptian and early agrarian societies showing seated or flexed working postures;

• Post-Industrial Period (Figures 4A–E, 5A–E, and 6A–E) – depictions of industrial laborers in factory and workshop settings.

All images were sourced from open-access or public-domain repositories (as detailed in the Figure Legends) and selected to illustrate common daily postures within each historical context.

The spinal alignments of the depicted figures were analyzed qualitatively, focusing on the curvature of the lumbar and thoracic regions, the degree of flexion or extension, and the nature of the depicted activity.

In Figure 1A–F, the observed postures predominantly show upright and dynamic movement patterns. In contrast, Figures 2A–F and 3A–F include numerous examples of forward flexion, seated work, and load handling with bent spines.

The industrial artworks (Figures 4A–E, 5A–E, and 6A–E) portray seated, leaning, or weight-bearing postures that imply sustained spinal loading.

This revision ensures a direct correspondence between the descriptive text and the associated visual materials, fully addressing the reviewer’s concern.

Reviewer Comment 3:

Discussion section: Some words are unnecessarily bolded; this formatting should be removed.

Response:

We have carefully reviewed the Discussion section and removed all unintended bold formatting. The text has been standardized for consistency and readability according to journal formatting guidelines.

We are grateful to the reviewer for their positive assessment and constructive feedback, which have helped us improve the precision and presentation quality of our manuscript.

Dear Scientist and Reviewer 2

We would like to express our sincere gratitude for your thoughtful and constructive feedback, which has significantly enhanced the quality, clarity, and scientific depth of our manuscript. Your insightful comments offered valuable perspectives that guided us in refining both the structure and the interpretation of our findings.

Throughout this process, we have learned a great deal from your expertise and critical insight, which has been truly enriching for us as researchers. Your guidance not only improved the precision of our arguments but also inspired us to approach the topic from a more comprehensive and balanced perspective.

We deeply appreciate the time and effort you devoted to reviewing our work and for helping us bring this study to a higher academic standard.

With warmest regards and sincere appreciation,

Abdullah Emre Taçyıldız on behalf of all co-authors

Reviewer Comment 1

Fascial and Myofascial Chain Perspective Consider referencing Thomas W. Myers’ “Anatomy Trains” model or Stecco’s fascial continuity research, which demonstrate how dysfunctions in one segment (e.g., thoracolumbar fascia, multifidus, quadratus lumborum) propagate tension throughout the kinetic chain. Chronic seated or flexed postures typically cause shortening of the hip flexors and thoracolumbar fascia, inhibition of the multifidus, and compensatory overactivity of the quadratus lumborum and upper trapezius—mechanically linking the pelvis, sacroiliac joint (SIJ), and shoulder girdle dysfunctions.

Response to Reviewer’s Comment 1:

As suggested, we have now clarified the functional integration of the spine within the global myofascial network by adding the following paragraph to the revised manuscript:

“Recent myofascial research, including the ‘Anatomy Trains’ model proposed by Myers and the fascial continuity studies of Stecco, emphasizes that the spine functions as part of an integrated kinetic chain [33–36]. Dysfunction in one segment—such as thoracolumbar fascial shortening or multifidus inhibition—may propagate tension through interconnected myofascial pathways, leading to compensatory overload in the pelvis, sacroiliac joint, and shoulder girdle [34–36]. These mechanisms reinforce our hypothesis that modern postural habits generate systemic biomechanical stress exceeding the spine’s evolutionary design (Figure 1–6).”

This paragraph has been incorporated into the revised version to address the reviewer’s suggestion and to strengthen the theoretical framework of the manuscript.

Reviewer Comment 2

Trigger Points and Postural Overload Introduce the role of myofascial trigger points (TrPs) as a micro-level consequence of prolonged overload or static posture. Citing Travell & Simons could strengthen the pathophysiological basis: sustained seated posture → ischemic muscle zones → TrP activation in the multifidus and quadratus lumborum → referred pain mimicking spinal disorders. The addition of ultrasound (US)-based confirmation of TrPs and US-guided dry needling (US-DN) (Bubnov R., Ultrasound Med Biol, 2011; https://link.springer.com/article/10.1186/1878-5085-3-13) would provide a modern pathophysiological bridge between evolutionary mismatch and clinical manifestations of low back pain.

Response to Reviewer’s Comment 2

In response to the reviewer’s valuable suggestion, we have expanded the discussion of the micro-physiological mechanisms underlying postural overload. The following paragraph has been added to the revised manuscript:

“At a micro-physiological level, prolonged static or flexed postures induce ischemic and hypoxic zones within postural muscles such as the multifidus and quadratus lumborum, facilitating myofascial trigger point (TrP) formation [37]. These TrPs can produce referred pain patterns closely resembling spinal pathology, thereby blurring the clinical distinction between functional and structural disorders. Recent ultrasound-based studies further confirm the presence of TrPs and demonstrate the therapeutic role of US-guided dry needling (US-DN) in reducing neuromyofascial hyperactivity [38, 39]. Together, these findings provide a mechanistic bridge linking evolutionary postural mismatch to the clinical manifestations of chronic low back pain (Figure 1–6).”

This paragraph was incorporated to strengthen the physiological and clinical correlation between modern postural behavior and spinal evolution, as recommended.

Reviewer Comment 3:

Comparative Functional Anatomy

The authors could contrast human static sitting posture with that of non-human primates (e.g., chimpanzees), which maintain a flexed hip and mobile lumbar spine position even when seated (see https://doi.org/10.1016/j.applanim.2021.105417;
https://doi.org/10.1016/0163-6383(91)90022-K). This comparison would show that humans’ prolonged sitting with posterior pelvic tilt and loss of lumbar lordosis is not “evolutionarily optimized,” unlike dynamic squat resting seen in apes.

Response to Reviewer’s Comment 3:

To further strengthen the comparative anatomical perspective as suggested, we have added the following paragraph to the revised manuscript:

“From a comparative anatomical standpoint, non-human primates such as chimpanzees maintain a flexed hip and dynamically mobile lumbar spine even when seated, preserving segmental motion and balanced load transmission across the pelvis and spine [40–43]. In contrast, the modern human sitting posture—marked by posterior pelvic tilt, reduced lumbar lordosis, and static hip flexion—deviates from these adaptive resting behaviors. This suggests that human prolonged sitting is not an evolutionarily optimized posture but rather a biomechanically compromised one emerging from cultural and occupational adaptations.”

This addition reinforces the evolutionary framework of the manuscript by contrasting human postural mechanics with those of non-human primates.

Reviewer Comment 4:

The authors might briefly acknowledge modern behavioral adaptations—such as the slightly head-down posture that improves social perception but may alter cervical–thoracic load distribution (see https://oa.mg/work/10.1016/j.actpsy.2022.103602)—to highlight ongoing tension between evolutionary optimization and modern psychosocial demands.

Response to Reviewer’s Comment 4:

In accordance with the reviewer’s insightful comment, we have enriched the discussion by integrating recent behavioral perspectives that link modern social habits to postural adaptation. The following paragraph has been added to the revised version:

“Recent behavioral studies indicate that modern humans often adopt a slightly head-down posture to enhance social perception and empathy cues in interpersonal communication [44]. While this adaptation may serve psychosocial purposes, it simultaneously increases cervical and upper thoracic mechanical load, illustrating the continuing conflict between evolutionary optimization and the postural demands of modern social behavior.”

This addition provides a behavioral and psychosocial dimension to the discussion, emphasizing how contemporary communication patterns may contribute to cervical and thoracic strain within an evolutionary context.

Dear Scientist and Reviewer 3

We sincerely appreciate your valuable and insightful comments, which have greatly contributed to improving the quality and clarity of our manuscript. We carefully reviewed each of your suggestions and have made the corresponding revisions accordingly. Your thoughtful feedback not only enhanced the scientific rigor of our work but also helped us present our findings in a more precise and balanced manner.

We are truly grateful for the time and expertise you dedicated to our study and for your constructive guidance throughout the review process.

With our deepest respect and appreciation,

Abdullah Emre Taçyıldız on behalf of all co-authors

Scientist and Reviewer 3, Comment 1:

This study provided an evidence-based alert regarding biomechanically unsustainable modern habits. However, I think there need add some scientific data in this article. The results show different positions were associated with intradiscal pressures. If the authors provided some data of intradiscal pressure under different position. These data can help to clearly tell the reader which position were risks of low back pain.

Response 1:

Based on your valuable suggestions, the following paragraph has been added to the manuscript.

According to the in vivo measurements reported by Wilke et al., intradiscal pressure varies substantially with body posture and physical activity, reflecting the dynamic loading characteristics of the lumbar spine. In a lying supine position, mean disc pressure remains minimal at approximately 0.10 MPa, increasing slightly when lying on the side (0.12 MPa) or prone (0.11 MPa) [13]. When the trunk is extended while prone and supported on the elbows, the pressure rises markedly to about 0.25 MPa, illustrating the load transfer through posterior spinal elements. Everyday actions such as laughing or sneezing in a lateral decubitus position elevate pressures to 0.15 MPa and 0.38 MPa, respectively, while turning movements can transiently peak at 0.70–0.80 MPa. During upright posture, relaxed standing produces around 0.50 MPa, which increases to nearly 1.10 MPa when bending forward, and further to 0.92 MPa during a Valsalva maneuver. Sitting generates variable loading depending on posture: from 0.27 MPa in a slouched position to 0.83 MPa with maximal flexion, while active upright sitting reaches approximately 0.55 MPa. Transitioning from sitting to standing momentarily produces high pressures near 1.10 MPa. Locomotor and lifting activities exert even greater variations. Walking yields 0.53–0.65 MPa, whereas jogging ranges from 0.35 to 0.95 MPa depending on footwear. Ascending or descending stairs causes pressures between 0.30 and 1.20 MPa. Notably, lifting 20 kg with a rounded back can increase intradiscal pressure to 2.30 MPa, compared to 1.70 MPa when proper lifting technique is employed, and 1.10 MPa when the load is held close to the body [13]. Maintaining the same load 60 cm from the chest produces 1.80 MPa, underscoring the mechanical disadvantage of lever arm extension. Even during nocturnal rest, pressures fluctuate between 0.10 and 0.24 MPa, indicating continuous low-level mechanical activity of the spine during recumbency. These data collectively highlight the remarkable sensitivity of spinal load distribution to posture, movement, and external weight, providing an essential biomechanical foundation for ergonomic recommendations and preventive strategies against lumbar disc degeneration.

Scientist and Reviewer 3, Comment 2:

The figure 2 provided paintings from ancient Egypt. But people in some pictures were not workers. This could not show the people’s daily tasks. Although the arts from ancient Egyptian are few. I still suggest the author provide more pictures of workers. In figure 1-6, please label alphabetic markers in subfigures, like A, B, C, et al.

Response 2:

We sincerely thank the reviewer for the valuable and insightful comments. In response, all figures have now been labeled alphabetically (A, B, C, etc.) for clarity, as suggested.

Regarding the concern about the inclusion of certain ancient Egyptian paintings in Figure 2, we fully understand and appreciate the reviewer’s observation. Our intention was not solely to depict laborers, but to illustrate a broader range of postural behaviors observed in ancient societies. The inclusion of seated individuals was deliberate, as we aimed to emphasize that the act of sitting itself—often portrayed as static and hierarchically symbolic—can represent an early form of non-ergonomic, and even potentially pathological, posture.

Nevertheless, we have revised the figure selection and organization to better align with the reviewer’s perspective. The images are now presented alphabetically and have been carefully chosen to maintain scientific and cultural relevance. We hope this rationale will be understood and accepted with appreciation.

Scientist and Reviewer 3, Comment 3:

In ancient records, it limited historical documentation on low back pain, which inferred its association with poor posture through artistic depictions appears methodologically indirect and lacks substantive foundation.

Response 3:

We appreciate the reviewer’s insightful comment. In response, we have added a statement to the Limitations section acknowledging that a key limitation lies in the scarcity of ancient records explicitly describing low back pain, which makes any inference about its link to poor posture based on artistic representations methodologically indirect and lacking robust evidential support.

Based on your valuable suggestions, the following paragraph has been added to the manuscript.

A key limitation lies in the scarcity of ancient records explicitly describing low back pain, making any inference about its link to poor posture based on artistic representations methodologically indirect and lacking robust evidential support.

Scientist and Reviewer 3,

---

## [Decision Letter · Decision Letter 1]

12 Nov 2025

Dear Dr. Abdullah Emre Taçyıldız,

Thank you for submitting your manuscript to PLOS ONE. After careful consideration, we feel that it has merit but does not fully meet PLOS ONE’s publication criteria as it currently stands. Therefore, we invite you to submit a revised version of the manuscript that addresses the points raised during the review process.

We look forward to receiving your revised manuscript.

Kind regards,

Holakoo Mohsenifar

Academic Editor

PLOS ONE

Journal Requirements:

Reviewers' comments:

Reviewer's Responses to Questions

**Comments to the Author**

Reviewer #1: All comments have been addressed

Reviewer #2: (No Response)

2. Is the manuscript technically sound, and do the data support the conclusions?

Reviewer #1: Yes

Reviewer #2: Yes

3. Has the statistical analysis been performed appropriately and rigorously?

Reviewer #1: N/A

Reviewer #2: N/A

4. Have the authors made all data underlying the findings in their manuscript fully available?

Reviewer #1: Yes

Reviewer #2: Yes

5. Is the manuscript presented in an intelligible fashion and written in standard English?

Reviewer #1: Yes

Reviewer #2: Yes

Reviewer #1: (No Response)

Reviewer #2: The revised manuscript presents an original and engaging concept linking spinal evolution, posture, and low back pain, supported by rich visual material. The authors have improved structure and contextual clarity and partially integrated functional anatomy and cultural aspects as suggested in the previous round.

However, some issues remain:

Conceptual Balance

While the manuscript argues that civilization and modern posture negatively affect spinal health, the examples introduced (e.g., primate posture, chimpanzee lumbar mobility, and cultural balancing practices such as head-load carrying in African populations) actually demonstrate that posture adaptation is multifactorial and not always detrimental. This complexity slightly contradicts the paper’s main narrative and should be acknowledged briefly in the discussion.

Functional and Clinical Integration

To strengthen translational value, the authors could still expand on the functional anatomy aspects—particularly the role of the thoracolumbar fascia, multifidus, quadratus lumborum, and pelvic chain—and how chronic overload leads to myofascial dysfunction.

Reference to treatments based on understanding biomechanics via myofascial trigger points, ultrasound diagnostics, and US-guided dry needling (US-DN) would help illustrate how the evolutionary–postural concept translates into clinical understanding and intervention.

Practical Perspective

The discussion would benefit from a short preventive and rehabilitative outlook, referring to dynamic posture training, fascial stretching, or core stabilization, US-DN, to connect evolutionary theory with clinical application.

Overall, this is an interesting and creative paper, presenting a thought-provoking view that stimulates interdisciplinary discussion. It could be further refined indefinitely, but the current version already makes a valuable conceptual contribution.

**Do you want your identity to be public for this peer review?** For information about this choice, including consent withdrawal, please see our Privacy Policy

Reviewer #1: No

Reviewer #2: **Yes:**  Rostyslav Bubnov

---

## [Author Response · Author response to Decision Letter 2]

14 Nov 2025

Dear Scientist and Reviewer

We would like to sincerely thank the reviewer not only for helping us improve the overall quality and clarity of our manuscript, but also for providing constructive insights from which we have learned a great deal scientifically. Their thoughtful and formative feedback has significantly enriched both the intellectual depth and the scientific rigor of our work.

Reviewer Comment 1: Conceptual Balance

While the manuscript argues that civilization and modern posture negatively affect spinal health, the examples introduced (e.g., primate posture, chimpanzee lumbar mobility, and cultural balancing practices such as head-load carrying in African populations) actually demonstrate that posture adaptation is multifactorial and not always detrimental. This complexity slightly contradicts the paper’s main narrative and should be acknowledged briefly in the discussion.

Author response 1:

We sincerely thank the reviewer for this insightful comment highlighting the need for conceptual balance. We fully agree that postural evolution is not a unidirectional process of degeneration but rather a multifactorial adaptation shaped by evolutionary, environmental, and cultural influences.

Accordingly, we have revised the relevant section of the discussion to clarify that the cited examples—such as habitual head-load carrying in African populations and lumbar flexibility in non-human primates—do not contradict our central argument. Instead, they contextualize modern spinal vulnerability as a mismatch between our evolved spinal morphology and the rapid biomechanical and behavioral changes brought by modern civilization.

This conceptual refinement strengthens the manuscript by integrating the reviewer’s valuable perspective and emphasizing that spinal health reflects both adaptive and maladaptive postural mechanisms operating along an evolutionary continuum. The following paragraph has been added accordingly.

While our central argument emphasizes that modern civilization has introduced postural challenges through sedentary behavior, artificial ergonomics, and diminished proprioceptive feedback, it is equally important to recognize that postural evolution itself is not a unidirectional process of degeneration. Anthropological and biomechanical research shows that posture has historically been a multifactorial adaptive trait shaped by environmental and cultural demands. For instance, studies of traditional African populations (Luo tribe) practicing habitual head-load carrying demonstrated reduced energetic cost and highly efficient load transfer during gait, reflecting adaptive strengthening of spinal and paraspinal structures [55, 56]. Women of the Kikuyu tribe, in contrast, carry substantial loads supported by a forehead strap, a practice that frequently leads to the development of a permanent cranial groove [55]. Likewise, comparative anatomical analyses indicate that non-human primates, including chimpanzees, exhibit distinct lumbar and pelvic adaptations that provide context-specific mobility and axial function, underscoring that spinal form varies according to ecological pressures [6]. These findings refine our argument by showing that posture has always adapted to its mechanical environment; thus, modern civilization may represent a mismatch between our evolved spinal design and the novel loading patterns of contemporary life. In this view, the issue is not evolutionary failure, but the rapid divergence between ancestral biomechanical demands and present-day postural exposures.

Reviewer Comment 2. Functional and Clinical Integration

To strengthen translational value, the authors could still expand on the functional anatomy aspects—particularly the role of the thoracolumbar fascia, multifidus, quadratus lumborum, and pelvic chain—and how chronic overload leads to myofascial dysfunction. Reference to treatments based on understanding biomechanics via myofascial trigger points, ultrasound diagnostics, and US-guided dry needling (US-DN) would help illustrate how the evolutionary–postural concept translates into clinical understanding and intervention.

Author Response 2:

In response to the reviewer’s valuable suggestion to enhance the translational relevance of the manuscript, we expanded the discussion on functional anatomy—specifically addressing the biomechanical roles of the thoracolumbar fascia, multifidus, quadratus lumborum, and the pelvic myofascial chain—and clarified how chronic asymmetric loading contributes to myofascial dysfunction. In addition, we integrated a brief overview of clinically relevant interventions grounded in biomechanical understanding, including the significance of myofascial trigger points, the role of ultrasound-based diagnostic approaches, and the therapeutic use of ultrasound-guided dry needling (US-DN). These revisions aim to more clearly bridge the evolutionary–postural framework with practical clinical applications. The following paragraph has been added accordingly.

Ultrasound (US) imaging has increasingly become a critical tool in the diagnosis and treatment of myofascial pain. Bubnov first demonstrated the feasibility of real-time sonographic visualization of myofascial trigger points and reported 93.3% pain relief in a clinical series of 91 patients treated with US-guided trigger-point dry needling (US-DN) [42]. Building on these findings, a comparative clinical study showed that US-guided needling provides more accurate localization, greater pain reduction, and higher treatment responsiveness than palpation-guided techniques [41]. More recent reports further suggest that US-guided interventions may enhance muscle function and regional mobility by enabling targeted treatment of fascial dysfunction [43]. Collectively, these data underscore how image-guided approaches translate myofascial biomechanics into precise clinical interventions, reinforcing the connection between fascial anatomy, postural evolution, and modern rehabilitation strategies. These ultrasound-guided myofascial findings support our central hypothesis by showing that the fascial and segmental dysfunctions produced by modern, evolution-mismatched loading patterns are not only measurable but clinically reversible. Thus, US-based interventions provide mechanistic confirmation that contemporary postural strain exceeds the adaptive conditions under which the human spine evolved.

From a practical standpoint, the evolutionary–postural framework highlights the need for interventions that restore dynamic spinal loading and myofascial mobility. Approaches such as dynamic posture training, fascial stretching, and core stabilization can counteract the sedentary, asymmetric loading patterns characteristic of modern environments [36]. In addition, ultrasound-guided dry needling (US-DN) provides a targeted method for reducing fascial densification and improving segmental control [40, 41]. Together, these strategies translate evolutionary principles into clinically actionable rehabilitation pathways.

Reviewer Comment 3: Practical Perspective

The discussion would benefit from a short preventive and rehabilitative outlook, referring to dynamic posture training, fascial stretching, or core stabilization, US-DN, to connect evolutionary theory with clinical application.

Author Response 3:

In accordance with the reviewer’s insightful recommendation to incorporate a concise preventive and rehabilitative perspective, we expanded the discussion to highlight clinically relevant strategies such as dynamic posture training, fascial stretching, core stabilization, and the complementary use of ultrasound-guided dry needling (US-DN). These additions were made to strengthen the bridge between the evolutionary framework and practical clinical management. The paragraph below has been added to the revised manuscript to address this point.

From a functional and clinical standpoint, the evolution of human posture cannot be fully understood without integrating the role of the myofascial system. The thoracolumbar fascia (TLF) forms a continuous tension network linking the spine, pelvis, and lower limbs, providing mechanical stability and efficient force transmission during movement [36, 37]. Within the evolutionary context of human bipedalism, the thoracolumbar fascia (TLF) functions as a key connective interface linking the trunk and pelvis, contributing to load transfer along the axial chain and stabilizing the spine during upright locomotion. In this framework, Schleip et al. further demonstrated that fascia is not a passive membrane but a contractile tissue: human lumbar fascia contains α-SMA–positive myofibroblasts capable of generating active tension, and subtle stiffness changes in these fascial layers may influence neuromuscular coordination through mechanosensory pathways, thereby integrating with the broader biomechanical adaptations that support human spinal posture [38]. Together, these findings support our central hypothesis that modern postural problems stem less from skeletal alignment alone and more from maladaptive changes within the myofascial system. If the thoracolumbar fascia and its paraspinal–pelvic networks function as contractile, load-responsive tissues prone to densification, stiffness shifts, and neuromuscular modulation, then the sedentary and asymmetric loading patterns of contemporary life may disrupt a system that once stabilized the trunk and coordinated lower-limb biomechanics. In this sense, modern postural dysfunction reflects an evolutionary mismatch between a formerly dynamic fascial architecture and today’s mechanically constrained environments.

---

## [Decision Letter · Decision Letter 2]

1 Dec 2025

Viewing low back pain through the lens of spinal evolution: Understanding the morphology and limits of the human spine

PONE-D-25-38294R2

Dear Dr. Abdullah Emre Taçyıldız

We’re pleased to inform you that your manuscript has been judged scientifically suitable for publication and will be formally accepted for publication once it meets all outstanding technical requirements.

Kind regards,

Holakoo Mohsenifar

Academic Editor

PLOS ONE

Additional Editor Comments (optional):

Reviewers' comments:

Reviewer's Responses to Questions

**Comments to the Author**

Reviewer #1: All comments have been addressed

Reviewer #2: All comments have been addressed

2. Is the manuscript technically sound, and do the data support the conclusions?

Reviewer #1: Yes

Reviewer #2: Yes

3. Has the statistical analysis been performed appropriately and rigorously?

Reviewer #1: N/A

Reviewer #2: N/A

4. Have the authors made all data underlying the findings in their manuscript fully available?

Reviewer #1: Yes

Reviewer #2: Yes

5. Is the manuscript presented in an intelligible fashion and written in standard English?

Reviewer #1: Yes

Reviewer #2: Yes

Reviewer #1: (No Response)

Reviewer #2: The authors have responded thoroughly and thoughtfully to all previous comments. The revised manuscript now demonstrates a substantially improved conceptual balance, clearer integration of comparative anatomy, and an enhanced connection between evolutionary posture, myofascial physiology, and clinical relevance.

The additions addressing:

evolutionary mismatch vs cultural/environmental adaptation,

the thoracolumbar fascia, pelvic chain, and functional anatomy,

myofascial trigger points diagnostics/treatment,

preventive and rehabilitative strategies,

are comprehensive and appropriate. The new discussion paragraphs effectively resolve the concerns raised earlier, especially regarding the need for conceptual nuance and translational context.

Importantly, the authors have succeeded in clarifying that examples such as chimpanzee lumbar mobility or African head-load carrying do not contradict their thesis, but rather illustrate the complexity of spinal adaptation and help frame the idea of evolutionary–environmental mismatch in modern sedentary contexts. This refinement strengthens the manuscript and removes the prior conceptual tension.

The clinical translation component (myofascial dysfunction, dynamic posture, fascial mobility) is now clearly articulated and adds meaningful value without overwhelming the conceptual nature of the paper.

Overall, the manuscript is now cohesive, scientifically sound, and intellectually stimulating.

No further major issues remain, and the paper could be revised indefinitely without essential improvement.

Recommendation: Accept

**Do you want your identity to be public for this peer review?** For information about this choice, including consent withdrawal, please see our Privacy Policy

Reviewer #1: No

Reviewer #2: **Yes:**  Rostyslav Bubnov

---

## [Editor Report · Acceptance letter]

PONE-D-25-38294R2

PLOS One

Dear Dr. Taçyıldız,

I'm pleased to inform you that your manuscript has been deemed suitable for publication in PLOS One. Congratulations! Your manuscript is now being handed over to our production team.

Kind regards,

on behalf of

Dr. Holakoo Mohsenifar

Academic Editor

PLOS One